# Natural cycle versus hormone replacement therapy as endometrial preparation in ovulatory women undergoing frozen-thawed embryo transfer: The COMPETE open-label randomized controlled trial

Xitong Liu[1], Wentao Li[2], Wen Wen[1], Ting Wang[1], Tao Wang[1], Ting Sun[1], Na Zhang[1], Dan Pan[1], Jinlin Xie[1], Xiaojuan Liu[1], He Cai[1], Xiaofang Li[1], Zan Shi[1], Rui Wang[1], Na Lu[1], Haiyan Bai[1], Rong Pan[1], Li Tian[1], Bin Meng[1], Xin Mu[1], Hongran Jia[1], Hanying Zhou[1], Xu Cao[3], Tianxing Liu[1], Pengfei Qu[4], Danmeng Liu[4], Ben W. Mol[5,6], Juanzi Shi[1]*

1 The Assisted Reproduction Center, Northwest Women's and Children's Hospital, Xi'an, China,
2 National Perinatal Epidemiology and Statistics Unit, Centre for Big Data Research in Health and School of Women's and Children's Health, The University of New South Wales, Sydney, Australia, 3 Graduate Department, Xi'an Medical University, Xi'an, China, 4 Translational Medicine Center, Northwest Women's and Children's Hospital, Xi'an, China, 5 Department of Obstetrics and Gynaecology, Monash University, Monash Medical Centre, Wellington Road, Clayton, Victoria, Australia, 6 Department of Obstetrics and Gynaecology, Amsterdam University Medical Centre, Amsterdam, The Netherlands

☯ These authors contributed equally to this work.
* shijuanzi123@126.com

## Abstract

### Background

Different endometrial preparation protocols are used prior to frozen-thawed embryo transfer (FET). Optimization of endometrial preparation protocols is mandatory to improve live birth rate and obstetric and perinatal outcomes. In the Comparison of Endometrial Preparation Protocols for Frozen Embryo Transfer (COMPETE) trial, our primary objective is to evaluate whether natural cycles (NCs) lead to a higher live birth rate after the first FET cycle compared to hormone replacement therapy (HRT) cycles in women with a regular ovulatory cycle.

### Methods and findings

We performed a single-center, parallel, open-label randomized controlled trial between December 2020 and December 2022 in a single assisted reproduction center in Xi'an, China. Women with a regular menstrual cycle undergoing in vitro fertilization (IVF) scheduled for FET were randomly assigned (1:1) to endometrial preparation in the NC or with HRT, using a web-based electronic data capture system. The primary outcome was live birth rate after the initial FET. The analysis was conducted based on the intention-to-treat principle. Obstetric and perinatal outcomes in all

**Data availability statement:** All relevant data from the COMPETE study are within the manuscript and its Supporting Information files. Data analysis code is available at https://osf.io/ym7sw/files/osfstorage?view_only=8077b-13da49d430e9d8145aa94de7fca

**Funding:** This work was supported by National Key Clinical Specialty Construction Project(to JS), Key Research and Development Program of Shaanxi (No. 2023-ZDLSF-48 to JS), the General Projects of Social Development in Shaanxi Province (No. 2022SF-565 to XTL), and the Xi'an Municipal Science and Technology Bureau, China under grants (24YXYJ0175 to XTL). The funders had no role in study design, data collection and analysis, decision to publish, or preparation of the manuscript.

**Competing interests:** I have read the journal's policy and the authors of this manuscript have the following competing interests: BWM reports consultancy, travel support and research funding from Merck and consultancy for Ferring, Organon, and Norgine. WL has received NHMRC Investigator grant (GNT2016729).

**Abbreviations:** FET, frozen-thawed embryo transfer; CI, confidence interval; COMPETE, the comparison of endometrial preparation protocols for frozen embryo transfer; EDC, Electronic data collection system; ICM, inner cell mass; IQR, Inter-quantile range; GDM, gestational diabetes mellitus; hCG, human chorionic gonadotropin; HPO, hypothalamic-pituitary-ovarian; HRT, hormone replacement therapy; IVF, in vitro fertilization; LGA, large for gestational age; LH, luteinizing hormone; NC, natural cycle; NICU, neonatal care intensive unit; PPROM, preterm premature rupture of membranes; RCT, randomized controlled trials; RD, risk difference; RR, risk ratio; SAP, statistical analysis plan;SGA, small for gestational age; TE, trophectoderm.

randomly assigned women were reported in this study. We randomly assigned 902 women to receive either NC ($n = 448$) or HRT ($n = 454$). In the NC group, 101 women received HRT because of no ovulation, while in the HRT group, 29 women received NC because of spontaneous ovulation. The number of live births was 242 (54.0%) in the NC group versus 195 (43.0%) in the HRT group (absolute difference, 11.1 percentage points, 95% CI 4.6 to 17.5; risk ratio (RR) 1.26, 95% CI 1.10 to 1.44). Miscarriage rates (RR 0.61, 95% CI 0.41 to 0.89) and the antepartum hemorrhage rates (RR 0.63, 95%CI 0.42 to 0.93) were lower in the NC group, with other obstetric and perinatal outcomes not significantly different.

## Conclusions

In women with a regular menstrual cycle undergoing FET, a strategy starting with NC endometrial preparation results in higher live birth rates than endometrial preparation with HRT. However, the permitted cross-over between arms limits certainty in directly assessing NC versus HRT efficacy.

## Trial registration

Chinese Clinical Trial Registry: ChiCTR2000040640.

## Author summary

### Why was this study done?

- The preparation of the endometrium is a crucial step in securing the success of frozen embryo transfer.

- Until recently, no well-designed RCTs had compared natural cycle with hormone replacement therapy, two of the most widely used protocols, for live birth rates and obstetric and perinatal outcomes in women with regular menstrual cycle with sufficient power.

- The optimal endometrial protocol for frozen embryo transfer is undetermined

### What did the researchers do and find?

- The COMPETE study is a large randomized controlled trial investigating clinical outcomes and complications between natural cycle and hormone replacement therapy in women with regular menstrual cycle scheduled for frozen embryo transfer.

- This study showed natural cycle for endometrial preparation led to a higher live birth rate, and a lower risk of miscarriage and antepartum hemorrhage, compared to hormone replacement therapy cycle.

**What do these findings mean?**

- Hormone replacement treatment should not be prioritized in women with regular menstrual cycle undergoing FET as it is associated with lower live birth rate and potentially higher risks of obstetric and perinatal complications including miscarriage rates and antepartum hemorrhage rates compared with natural cycle.

- Natural cycle appears to be the preferred protocol for ovulatory women undergoing their first frozen embryo transfer treatment.

- Limitations of this study include its single-center design, the allowance for cross-over between arms under specific conditions, and insufficient statistical power to assess rare pregnancy complications.

## Introduction

In vitro fertilization (IVF) is the cornerstone of modern infertility treatment. The utilization of frozen-thawed embryo transfer (FET) in IVF has significantly risen in recent decades. This approach minimizes surplus embryo wastage, mitigates the risk of ovarian hyperstimulation syndrome, enhances cumulative live birth rates, particularly in patients experiencing hyperstimulation, enables preimplantation genetic testing for aneuploidy (PGT-A), supports fertility preservation, and provides flexibility in scheduling embryo transfer [1]. Furthermore, frozen embryos are transferred into a more physiologic non-stimulated endometrium which could potentially reduce maternal and neonatal complications [1].

In the preparation of frozen embryo transfer, different protocols have been employed to synchronize the development of the embryo and the endometrium. Among these protocols, the natural cycle (NC) and hormone replacement therapy (HRT) are two commonly used methods for endometrial preparation. A recent Cochrane review concluded that there is insufficient evidence on the priority of endometrial preparation protocol [2]. Our previous retrospective cohort study in young women with regular menstrual cycles reported higher live birth rates after NC than HRT [3]. In addition, observational studies have suggested that HRT cycles are associated with an increased risk of obstetric and neonatal complications compared to NC [4,5]. This phenomenon may be attributed to the absence of the corpus luteum and consequent reduced secretion of vasoactive substances like vascular endothelial growth factor and relaxin in HRT cycles, or to endometrial changes caused by HRT [6].

In view of increasing prevalence of FET cycles worldwide and the ongoing debate regarding different endometrial preparation protocols, we conducted a randomized controlled trial (RCT) comparing endometrial preparation with NC and HRT in women with regular menstrual cycle undergoing FET after IVF.

## Methods

This study was reported according to the Consolidated Standards of Reporting Trials (CONSORT) Statement (see S2 Text).

### Study design

The comparison of endometrial preparation protocols for frozen embryo transfer (COMPETE) was a single-center, open-label, parallel-group, randomized controlled trial conducted at the Assisted Reproduction Center, Northwest Women's and Children's Hospital, Xi'an, China. Participants were randomized in a 1:1 ratio to 2 study arms to compare different endometrial preparation protocols for frozen embryo transfer: NC arm and HRT arm. The study was registered at the Chinese clinical trial registry (www.chictr.org.cn) as ChiCTR2000040640 on December 05, 2020, and was approved by the ethics committee of Northwest Women's and Children's Hospital (No. 2020008). The trial was monitored by an independent external Data and Safety Monitoring Board (DSMB) with clinical and statistical expertise (members at the end of the manuscript). The trial was conducted in accordance with the Declaration of Helsinki and the local laws of relevant

regulatory authorities. Participant recruitment, counseling, follow up, data-entry, and management were conducted at Northwest Women's and Children's Hospital. This trial was funded by the National Key Clinical Specialty Construction Project, Key Research and Development Program of Shaanxi, the General Projects of Social Development in Shaanxi Province, and the Xi'an Municipal Science and Technology Bureau, without any additional commercial support. The trial protocol, which has been previously published [7], can be accessed in the protocol, available as S3 Text. The first and corresponding authors assume responsibility for the accuracy and completeness of the data.

Infertile individuals attending the recruitment center between December 15, 2020, and December 20, 2022 with a regular menstrual cycle scheduled for FET after IVF were eligible. A regular menstrual cycle was defined as a cycle length between 21 and 35 days. Women with ovulation disorders, defined as irregular ovulation or anovulation, were not eligible, as were women with intrauterine adhesions diagnosed at hysterosalpingography, hysteroscopy, or transvaginal ultrasound, which are part of the routine checks conducted prior to IVF treatment. Due to regulatory requirements in China, only married women are permitted to undergo IVF treatment. This includes women with conditions such as gamete transport obstruction, cervical factors, ovulatory disorders, endometriosis, and unexplained infertility. Additionally, baby sex selection and surrogacy are prohibited. Overall, the efficacy and safety indicators of IVF in China are comparable to those in Western countries [8].

### Randomization

Before the start of the FET treatment, eligible women were informed about the trial by a dedicated clinical research team, with 48 h to consider participation. All participants gave written informed consent prior to randomization.

After obtaining informed consent, we randomly assigned women to either the NC protocol or the HRT protocol using simple randomization through a web-based electronic data capture system (ResMan, www.medresman.org.cn). This system includes a central randomization method that ensures allocation concealment. The randomization list was prepared by an independent statistician. A research assistant logs into the online system and inputs the participant's information into ResMan to determine their allocation. This allocation is recorded and cannot be changed. Randomization was done on menstrual cycle day 5. Given the open-label nature of the treatments, neither the doctors administering the interventions nor the participants themselves could be blinded to treatment allocation due to the inherent characteristics of the intervention. However, embryologists as well as physicians performing the embryo transfer were masked to group assignments.

### Intervention

NC or HRT cycles were initiated after the first menstrual period following oocyte retrieval and a negative pregnancy test. Participants assigned to the NC group were monitored through serial transvaginal ultrasound, starting on day 5 of the menstrual cycle. When the diameter of the dominant follicle reached 14 mm, serum luteinizing hormone (LH) was measured combined with transvaginal ultrasound daily. After confirming the detection of an LH surge (serum LH > 20 IU/L) with ultrasound evidence of collapsed follicles, FET could be scheduled. In situations where ultrasound imaging was unable to conclusively identify signs of ovulation, we relied on the detection of the LH surge to establish the timing for embryo transfer. In case an LH surge could not be detected (diameter of dominant follicle >17 mm while LH < 20 IU/L), 10,000 IU of urinary human chorionic gonadotropin (hCG) could be administered to trigger ovulation. For cleavage embryos, the optimal timing for FET was scheduled on ovulation + 3 day, LH surge + 4 day, or hCG + 5 day, while it was scheduled on ovulation + 5 day, LH surge + 6 day, or hCG + 7 day for blastocyst embryos. In case no dominant follicle growth was observed on day 10, women started with HRT using 6 mg oral estradiol valerate for 10 days. Luteal phase support of 200 mg vaginal micronized progesterone thrice daily was started from the day of ovulation.

In the study cohort assigned to the HRT group, a daily regimen of 6 mg oral estradiol valerate was initiated commencing from the fifth day of the menstrual cycle. After 5 days, an assessment of the endometrial thickness was conducted utilizing transvaginal ultrasound. Based on the results of this assessment, the dosage of estradiol could potentially be

escalated to a maximum of 8 mg per day. Once the endometrial thickness was confirmed to be favorable (at least 7 mm) and serum progesterone levels were less than 1.5 ng/ml after 10–12 days of estradiol administration, luteal phase support with 200 mg vaginal micronized progesterone thrice daily was started to schedule FET. Additionally, oral estradiol was continued with the same dosage. Women assigned to HRT who exhibited spontaneous follicle growth during endometrial preparation were monitored in a manner consistent with the NC group.

Dydrogesterone, at a dosage of 10 mg, was administered orally three times daily commencing from the day of embryo transfer. Oral estradiol was gradually diminished if clinical pregnancy was confirmed. Luteal phase support was continued until the 10th week of pregnancy.

In both the NC and the HRT group, embryo transfer could be canceled because of endometrial cavity fluid visible at ultrasound, a progesterone level >1.5 ng/ml on hCG trigger day, medical reasons, logistical constraints due to the COVID-19 pandemic, and personal non-medical reasons. In those cases, FET would be rescheduled.

Embryos were cultured in cleavage media after oocyte retrieval. The morphologic score was given on day 3 after oocyte retrieval according to the number of blastomeres, homogeneous degree of blastomeres, and degree of cytoplasmic fragmentation according to the Gardener's score [9]. Details of embryo scoring have been described elsewhere [10]. Briefly, embryos with 8–10 blastomeres, even homogeneous blastomeres <10% cytoplasmic fragmentation was graded as grade I, embryos with 6–7 or >10 blastomeres with even homogeneous blastomeres of no cytoplasmic fragmentation were graded as grade II, embryos with uneven and non-homogeneous blastomeres with 20%–50% cytoplasmic fragmentation were graded as grade III, and embryos with uneven and non-homogeneous blastomeres with >50% cytoplasmic fragmentation were graded as grade IV. Embryos graded I–III were considered usable embryos, while embryos graded I–II were regarded as good-quality embryos. For women with sufficient good-quality embryos (more than 4) on day 3, blastocyst culture would be performed. The grading system of blastocyst was evaluated according to the cell number and cell junction of both the inner cell mass (ICM) and trophectoderm (TE), as well as the expansion status based on the Gardener's score [9]. Blastocysts graded 4CC or higher were considered usable. Blastocysts with Gardner's A or B grades for ICM and TM were defined as good-quality blastocysts. The embryos were transferred in accordance with their ranking score. Elective single good quality blastocyst was transferred with prioritization. Otherwise, the stage of embryos in combination with morphology and quality were considered to select the embryos with high priority.

Embryos were preferred for fresh transfer first. In case of women with high progesterone levels on hCG trigger day (>1.5 ng/ml), high risk of ovarian hyperstimulation syndrome, or performing preimplantation genetic testing, a freeze-all policy was used. In instances where the fresh transfer cycle was unsuccessful, or if a live birth had occurred in the fresh cycle and a second child was desired, FET was conducted. These women were eligible for the present trial.

The cleavage embryos were warmed in the afternoon, one day prior to the embryo transfer, with a post-warm culture duration of 18–20 h. The blastocyst embryos were thawed on the same day as the embryo transfer.

Based on the patients' age, medical history, embryo stage, and quality, we transferred one or two embryos. Typically, women <37 years undergoing their first or second transfer cycle had single embryo transfer [11].

Serum β-hCG was measured 2 weeks after embryo transfer. Women with a positive pregnancy test were scheduled for ultrasound at 7 weeks gestation to confirm clinical pregnancy. Ultrasound was performed to confirm ongoing pregnancy at 12 weeks gestation in case of first ultrasound showed an intrauterine pregnancy.

## Outcomes

The primary outcome measure was live birth resulting from the first FET, which was defined as the delivery of a living infant(s) ≥28 weeks. We reported endometrial thickness, incidence of cycle cancelation, biochemical pregnancy, clinical pregnancy, miscarriage, ongoing pregnancy, and multiple pregnancy as prespecified secondary reproductive outcomes. Definitions of outcomes are provided in the Protocol, available as S1 Text.

For women with ongoing pregnancies, we recorded maternal and perinatal outcomes, all of which were predetermined. These outcomes included gestational diabetes mellitus (GDM), hypertensive disorders of pregnancy, antepartum hemorrhage, preterm birth, and various birth weight categories (low birth weight (defined as birth weight <2,500 g), very low birth weight (defined as birth weight <1,500 g), high birth weight (defined as birth weight >4,000,g), very high birth weight (defined as birth weight >4,500 g), large for gestational age (LGA), and small for gestational age (SGA)).

Antepartum hemorrhage was defined as bleeding in late pregnancy (>20 weeks gestation). Women who reported spotting or very light bleeding were not classified as experiencing antepartum hemorrhage.

Additionally, we recorded instances of congenital anomalies and perinatal mortality.

To further assess the effectiveness and safety of the treatment, we also reported several post hoc outcomes including ectopic pregnancy, maternal hyperthyroidism, maternal hypothyroidism, polyhydramnios, oligohydramnios, postpartum anemia, preterm premature rupture of membranes (PPROM), mode of delivery, gestational age at birth, and neonatal care intensive unit (NICU) admission.

### Follow-up of participants

Outcomes were collected through a standard clinical electronic data collection system (EDC) and inpatient records, at three time points: 7 weeks gestation, 12 weeks gestation, and completion of pregnancy. If the outcomes were not available from EDC or inpatients records, they were obtained through telephone contact or WeChat by dedicated research nurses. If a participant failed to respond following four separate attempts to establish contact at various times, they were deemed to have been lost to follow-up.

Adverse events were assessed at each participant visit, regardless of suspected causal relationship to medication. All participants were given access to 24 h emergency service and could report adverse events at any time.

### Sample size

According to the observational data of our center, the live birth rate of women <35 years old with regular menstrual cycle in the HRT group was 55.1% [3]. After consulting with gynecologists and epidemiologists, we considered a 10.1% absolute increase in live birth rate as clinically relevant. To be 80% certain that a two-sided 95% confidence interval could demonstrate or refute a 10.1% higher live birth rate, it was estimated that a sample size of 370 participants in each group would be necessary (740 participants in total). Assuming that 20% drop-out and protocol violation rate, we planned to randomize 888 participants.

### Statistical analysis

Analyses of the primary and secondary outcomes were done according to the intention-to-treat (ITT) principle. Baseline characteristics were described, and the balance between the two arms was assessed. For continuous variables, the normality test was estimated using frequency histograms and the Shapiro test initially. If the parameters were non-normally distributed, their medians and inter-quantile ranges (IQRs) were reported. The proportions of the two arms were presented for categorical variables. Risk ratios (RRs) and their 95% confidence intervals (95% CIs) for binary outcomes including singleton perinatal outcomes were compared using generalized lineal model with Poisson distribution and log link function and robust variance estimate. We chose robust Poisson regression models primarily because they are less sensitive to outliers and exhibit fewer convergence issues in comparison to log-binomial models. Continuous outcomes were compared using the independent samples *t* test after confirming normal distribution. For twin pregnancies, to account for possible nonindependence among neonates from the same pregnancy, we used generalized estimating equations (independent correlation structure) to consider the correlation between neonates in twin pregnancy. We used robust variance estimate to mitigate possible misspecification of the correlation structure. Women with missing outcomes were excluded from analyses of the affected outcomes. No imputation was made for missing values and no covariate adjustment was

used. A per-protocol analysis was performed in those who complied with the allocated treatment. We also performed a post hoc instrumental variable analysis for the primary outcome (random assignment as the instrument and actual treatment as the exposure) to understand the impact of cross-over between the study arms. Prespecified exploratory subgroup analyses included age group (<35 years *versus* ≥ 35 years), freeze-all policy in fresh cycles (yes *versus* no), and embryo stages (cleavage *versus* blastocyst). A *P*-value for interaction was computed by including an interaction term (subgroup Factor*treatment) in the generalized linear models to assess the treatment effect variation among different subgroups. Because there were a small number of women who had their second or third FET cycles randomized, we did a sensitivity analysis that only included first frozen transfer cycles. Statistical analyses were done using Stata 17.0. The significance level (*α*) is set at 0.05. A statistical analysis plan (SAP) was established on September 10, 2023, with the data becoming available to the investigators on September 17, 2023 (see S1 Text COMPETE trial statistical analysis plan). The SAP was finalized prior to data lock and independently of any knowledge of the data.

## Results

We screened 5,848 patients, of whom 2,989 had ovulation disorders, 991 had intrauterine adhesions, 749 refused to participate, and 217 were not approached due to administrative reasons. No age exclusion criteria were applied during the participant recruitment and enrollment process. After informed consent, 902 participants were randomly assigned to the NC (*n* = 448) or the HRT (*n* = 454) strategy for endometrial preparation (Fig 1). The last follow-up was completed in October 2023. Patient recruitment was concluded upon reaching the target sample size, resulting in a total of 902 patients being recruited, slightly exceeding the initially calculated sample size of 888. This occurred because multiple patients were simultaneously undergoing the screening process at the recruitment's conclusion. In the NC group, 17 women had their second FET randomized following unsuccessful conception attempts during the initial FET. Similarly, in the HRT group, 14 women had second FET randomized and one woman had third FET randomized.

Among the 448 participants allocated to the NC group, 105 women experienced an absence of ovulation. Within this subset, 101 women received HRT group, while the remaining four were treated with GnRH agonist pretreatment prior to HRT due to clinical suspicions of adenomyosis. Seventeen women did not perform embryo transfer and one did not complete follow-up after live birth.

Of the 454 participants assigned to HRT group, 29 women had spontaneous ovulation during HRT cycle and therefore switched to NC, 12 women underwent mild stimulation, and 3 women had GnRH agonist pretreatment before HRT because of clinical suspicion of adenomyosis. Fourteen women did not perform embryo transfer, while one woman had three embryos transferred.

A total of 448 women were included in the NC group, and 454 women in the HRT group for the ITT analysis. In the per-protocol analysis, a total of 343 women were included in the NC group, while 410 women were included in the HRT group.

Baseline characteristics and clinical variables were balanced between the two groups (Table 1). The average age was 30.7 years in the NC group and 31.3 years in the HRT group, respectively. Tubal factor infertility was the most frequent indication (52.5% in NC versus 50.9% in HRT). Cycle characteristics between NC and HRT were also comparable in the two groups, with most women receiving single blastocyst transfer (84.9% in NC versus 80.4% in HRT) (Table 2). In the NC group, preimplantation genetic testing for aneuploidy (PGT-A) was performed in 14 out of 448 women (3.1%). In the HRT group, 13 out of 454 women (2.9%) underwent PGT-A. Most cycle cancelations were because of personal and non-medical requests or logistical constraints due to the COVID-19 pandemic.

In the ITT analysis, 242 (54.0%) women allocated to NC had live birth, compared to 195 (43.0%) live births in women allocated to HRT (risk difference (RD) 11.1%, 95% CI 4.6% to 18%; RR 1.26, 95% CI 1.10 to 1.44; Table 3).

For reproductive outcomes, participants in the NC had significantly thicker endometrium than in the HRT group (mean difference 0.57 mm, 95% CI 0.38 to 0.76 mm). Participants in the NC group had a higher incidence of biochemical

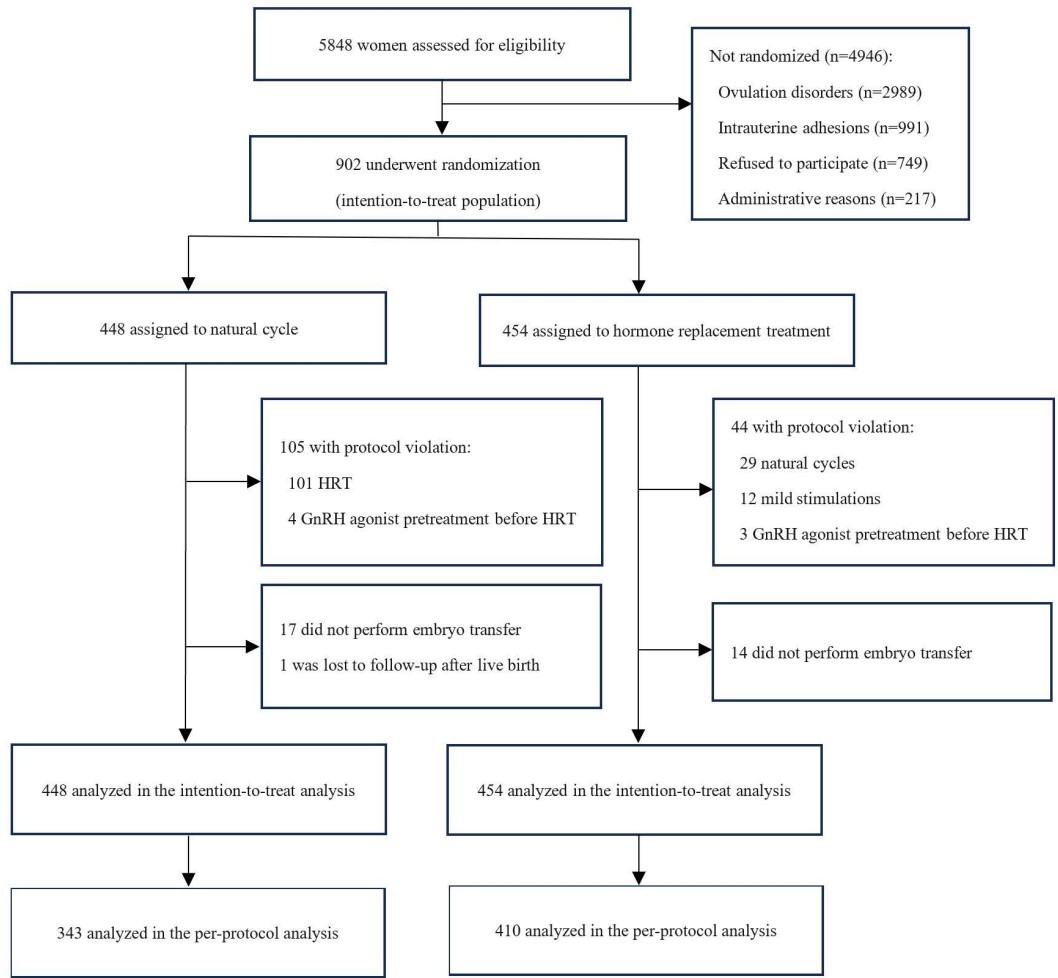

**Fig 1. Flowchart of study cohort.**

pregnancy (RD 8.6%, 95% CI 2.4% to 14.9%; RR 1.15, 95% CI 1.04 to 1.26), clinical pregnancy (RD 7.0%, 95% CI 0.6% to 13.4%; RR 1.12, 95%CI 1.01 to 1.25), and ongoing pregnancy (RD 11.1%, 95% CI 4.6% to 17.6%; RR 1.25, 95%CI 1.10 to 1.43). Participants in the NC group had a significantly lower rate of miscarriage (RD −8.4%, 95% CI −14.8% to −2.1%; RR 0.61, 95% CI 0.41 to 0.89) compared to participants who received HRT.

Regarding obstetric and perinatal outcomes, there was a significantly lower risk of antepartum hemorrhage in the NC group compared to the HRT group (RD −8.6%, 95% CI −16.0% to −1.2%; RR 0.63, 95% CI 0.42 to 0.93). Among 80 women who had antepartum hemorrhage, 15 had preterm births, 6 progressed to PPROM, while 5 had both preterm births and PPROM. The risks of GDM (10.9% versus 16.4%, RD −5.5%, 95% CI −12.0% to 0.9%; RR 0.66, 95%CI 0.41–1.06), hypertensive disorders of pregnancy (9.3% versus 8.0%, RD 1.3%, 95% CI −3.9% to 6.5%; RR 1.17, 95% CI 0.63 to 2.14), and preterm birth (8.9% versus 12.4%, RD −3.6%, 95% CI −9.3% to 2.2%; RR 0.71, 95%CI 0.41 to 1.23) were not statistically significantly different for NC versus HRT (Table 4).

Neonatal outcomes in terms of weight, large for gestational age, small for gestational age, and congenital anomaly were also not statistically significant different between the two groups. There were no cases of perinatal mortality in either group (Table 4).

**Table 1. Baseline and clinical characteristics[a].**

| | NC (*n* = 448) | HRT (*n* = 454) |
|---|---|---|
| Female age at freeze (y, mean, SD) | 30.73 (3.72) | 31.32 (4.28) |
| Female age at randomization (y, mean, SD) | 31.53 (3.59) | 32.02 (4.28) |
| Male age (y, mean, SD) | 32.00 (4.33) | 32.66 (4.83) |
| Infertility duration (y, median, IQR) | 3.00 (2.00, 4.00) | 3.00 (2.00, 5.00) |
| Female BMI (kg/m², median, IQR) | 22.33 (3.13) | 22.23 (3.04) |
| AFC (median, IQR) | 13.0 (10.00, 18.00) | 12.0 (8.00, 16.00) |
| Basal FSH (IU/L, median, IQR) | 6.70 (5.73, 7.98) | 6.74 (5.73, 8.39) |
| Infertility type | | |
| Primary infertility | 243 (54.2) | 223 (49.1) |
| Secondary infertility | 205 (45.8) | 231 (50.9) |
| Parity | | |
| Nulliparous | 385 (85.9) | 369 (81.3) |
| Multiparous | 63 (14.1) | 85 (18.7) |
| Infertility factors[b] | | |
| Tubal factor | 235 (52.5) | 231 (50.9) |
| Diminished ovarian reserve[c] | 12 (2.7) | 12 (2.6) |
| Endometriosis | 3 (0.7) | 2 (0.4) |
| Mixed factor[d] | 69 (15.4) | 82 (18.1) |
| Male factor | 29 (6.5) | 45 (9.9) |
| Other | 100 (22.3) | 82 (18.1) |
| Previous pregnancies | | |
| 0 | 252 (56.3) | 226 (49.8) |
| 1 | 103 (23.0) | 126 (27.8) |
| 2 | 51 (11.4) | 56 (12.3) |
| >2 | 42 (9.4) | 46 (10.1) |
| Abnormal thyroid stimulating hormone[e] | 50 (11.2) | 41 (9.0) |
| Protocol in fresh cycle | | |
| Agonist | 262 (58.5) | 256 (56.4) |
| Antagonist | 166 (37.1) | 175 (38.5) |
| Mild stimulation and natural cycle | 20 (4.5) | 23 (5.1) |
| Fertilization type | | |
| IVF | 335 (74.8) | 343 (75.6) |
| ICSI | 110 (24.6) | 108 (23.8) |
| IVF + ICSI | 3 (0.7) | 3 (0.7) |
| Gonadotropin dose (IU, median, IQR) | 2025.0 (1500.0, 2587.5) | 2025.0 (1575.0, 2600.0) |
| Gonadotropin duration (days, median, IQR) | 10.0 (9.0, 11.0) | 10.0 (9.0, 11.0) |
| No. of obtained oocytes (median, IQR) | 13.0 (9.0, 18.0) | 13.0 (8.0, 18.0) |
| No. of available embryos (median, IQR) | 7.0 (4.0, 10.0) | 6.0 (4.0, 9.0) |
| No. of good quality embryos (median, IQR) | 5.0 (2.0, 7.0) | 4.0 (2.0, 7.0) |
| Freeze-all policy | 317 (70.8) | 330 (72.7) |
| Previous fresh cycle without live birth | 99 (22.1) | 99 (21.8) |
| Live birth in previous fresh cycle and second child wish | 32 (7.1) | 25 (5.5) |

[a]Data are presented as *n* (%) unless stated otherwise. 2 missing for AFC; 2 missing for basal FSH; 5 missing for gonadotropin dose and duration; 6 missing for No. of good quality embryos.

[b]Shown with a maximum of one event per woman.

[c]Diminished ovarian reserve is marked by poor ovarian response in IVF, shown by high basal FSH (≥10 IU/L), low AMH (<1.1 ng/mL), and/or low AFC (<5~7).

*(Continued)*

**Table 1.** (Continued)

[d]Including patients with >1 female cause of infertility.

[e]Thyroid-stimulating hormone levels < 0.39 mIU/L or > 2.5 mIU/L. These women were already undergoing relevant medical treatment and were not recently diagnosed. Women exhibiting abnormal TSH levels were treated with levothyroxine to these levels before proceeding with IVF.

Abbreviations: NC, natural cycle; HRT, hormone replacement treatment; IQR, interquartile range; BMI, body mass index; AFC, antral follicle count; IVF, in vitro fertilization; ICSI, intracytoplasmic sperm injection.

**Table 2.** Characteristics of the natural cycles and HRT cycle.

|  | NC<br>n/N (%) | HRT<br>n/N (%) | P-value[a] |
|---|---|---|---|
| Reasons for cycle cancelation |  |  | 0.713 |
| Endometrial cavity fluid | 0/17 (0.0) | 2/14 (14.3) |  |
| High progesterone level | 1/17 (5.9) | 1/14 (7.1) |  |
| Medical reason (tuberculosis, illness) | 2/17 (11.8) | 2/14 (14.3) |  |
| COVID-19 pandemic | 4/17 (23.5) | 3/14 (21.4) |  |
| Personal and non-medical reason | 10/17 (58.8) | 6/14 (42.9) |  |
| Immediate transfer after oocyte retrieval | 132/448 (29.5) | 122/454 (26.9) | 0.387 |
| Category of natural cycle[b] |  |  | 0.449 |
| True NC | 225/343 (65.6) | 17/29 (58.6) |  |
| Modified NC | 118/343 (34.4) | 12/29 (41.4) |  |
| Estradiol dose increment to 8 mg per day in HRT[b] | 11/101 (10.9) | 57/410 (13.9) | 0.425 |
| Embryo stage |  |  | 0.082 |
| Cleavage (Day 3) | 65/431 (15.1) | 86/440 (19.6) |  |
| Blastocyst | 366/431 (84.9) | 354/440 (80.4) |  |
| Day5/Day6 blastocyst |  |  | 0.740 |
| D5 | 326/366 (89.1) | 318/354 (89.8) |  |
| D6 | 40/366 (10.9) | 36/354 (10.2) |  |
| PGT-A | 14/448 (3.1) | 13/454 (2.9) | 0.818 |
| No. of embryos transferred |  |  | 0.300 |
| 1 | 351/431 (81.4) | 370/440 (84.1) |  |
| ≥2 | 80/431 (18.6) | 70/440 (15.9) |  |
| No. of good-quality embryos transferred |  |  | 0.628 |
| 0 | 66/431 (15.3) | 78/440 (17.7) |  |
| 1 | 336/431 (78.0) | 334/440 (75.9) |  |
| 2 | 29/431 (6.7) | 28/440 (6.4) |  |

[a]Chi-squared test or Fisher's exact test.

[b]In the NC group, 105 women experienced an absence of ovulation, in which 101 women received HRT, four were treated with GnRH agonist pretreatment prior to HRT; In the HRT group, 29 women had spontaneous ovulation during HRT cycle and therefore switched to NC.

Abbreviations: NC, natural cycle; HRT, hormone replacement treatment; COVID-19, coronavirus disease 2019.

Post-hoc outcomes including ectopic pregnancy, maternal hyperthyroidism, maternal hypothyroidism, polyhydramnios, oligohydramnios, postpartum anemia, PPROM, mode of delivery, gestational age at birth, and NICU admission, revealed no significant differences between the NC and HRT groups (Table 4).

Overall, adverse events during the treatment period were reported in 3.1% of the participants in the HRT group and 3.4% of the participants in the NC group (S1 Table).

**Table 3. Reproductive outcomes (intention-to-treat analysis).**

| Clinical outcomes | NC | | HRT | | Absolute difference/mean difference (95% CI)[a] | Risk ratio (95% CI)[a] |
|---|---|---|---|---|---|---|
| | N | n (%)/mean (SD) | N | N (%)/mean (SD) | | |
| Live birth | 448 | 242 (54.0) | 454 | 195 (43.0) | 11.1 (4.6, 17.5) | 1.26 (1.10, 1.44) |
| Endometrial thickness (mm) | 439 | 10.9 (1.6) | 447 | 10.4 (1.3) | 0.57 (0.38, 0.76) | – |
| Cycle cancelation | 448 | 17 (3.8) | 454 | 14 (3.1) | 0.7 (−1.7, 3.1) | 1.23 (0.61, 2.47) |
| Biochemical pregnancy | 448 | 304 (67.9) | 454 | 269 (59.3) | 8.6 (2.4, 14.9) | 1.15 (1.04, 1.26) |
| Clinical pregnancy | 448 | 285 (63.6) | 454 | 257 (56.6) | 7 (0.6, 13.4) | 1.12 (1.01, 1.25) |
| Miscarriage | 285 | 37 (13.0) | 257 | 55 (21.4) | −8.4 (−14.8, −2.1) | 0.61 (0.41, 0.89) |
| Ongoing pregnancy | 448 | 248 (55.4) | 454 | 201 (44.3) | 11.1 (4.6, 17.6) | 1.25 (1.10, 1.43) |
| Multiple pregnancy | 448 | 18 (4.0) | 454 | 19 (4.2) | −0.2 (−2.8, 2.4) | 0.96 (0.51, 1.80) |
| Ectopic pregnancy* | 448 | 6 (1.3) | 454 | 7 (1.5) | −0.2 (−1.8, 1.4) | 0.87 (0.29, 2.56) |

[a]HRT group was regarded as the reference group.

*Post hoc specified endpoints.

Abbreviations: NC, natural cycle; HRT, hormone replacement treatment; CI, confidence interval.

Results for reproductive outcomes (S2 Table) and obstetric and perinatal outcomes (S3 Table) in the per-protocol population were similar to those in the main analysis. In the instrumental variable analysis, being randomized to NC increased the probability of receiving it by 70.6% (95% CI 66.1% to 75.2%). The F-statistic was 928.46, suggesting randomization strongly predicts treatment received. The treatment effect of NC versus HRT on live birth rate adjusted for cross-over was 15.6% (95% CI 6.4% to 24.8%).

No significant difference in treatment effects were found across prespecified subgroups with female age, freeze-all policy, and embryo stage in the intention-to-treat analysis (Fig 2). The sensitivity analyses of the reproductive outcomes (S4 Table) and maternal and perinatal outcomes (S5 Table), which only included first frozen-thawed embryo cycles, showed results that were similar to those of the primary analysis. The sensitivity analysis of reproductive outcomes (S6 Table), which included only those who had embryo transfer, produced results consistent with the main analysis.

## Discussion

In this randomized controlled trial, we found that the live birth rate was higher with a strategy starting with the NC protocol for endometrial preparation compared to HRT, in women undergoing FET with regular menstrual cycles. Additionally, a decreased miscarriage rate was noted in the NC group in contrast to the HRT group. Notably, factors such as female age, the implementation of a freeze-all policy during fresh cycles, and the stage of the embryo did not alter the treatment efficacy of NC in comparison to HRT.

Synchronization between endometrium and embryo development is a contributing factor for successful implantation. Though embryo quality plays a vital role in IVF, endometrial receptivity is also indispensable. The comparison between NC and HRT protocols, which are the two most commonly utilized methods for endometrial preparation, has been a subject of ongoing debate. In a NC approach, endometrium is built up by the endogenous estrogen produced by the growing follicle. However, monitoring of follicular growth or LH surge is required to determine the optimal timing of embryo transfer. In an HRT cycle, sequential treatment with estrogen and progesterone is administered to simulate the menstrual cycle, and this method is now commonly used because of its convenience. However, potential safety issues of HRT have been raised owing to the less physiological condition compared with NC.

Evidence before our study could not support one endometrial preparation being superior to another [12]. A 2019 RCT showed there were no significant differences in the live birth rates and other clinical outcomes between endometrial preparation methods in women with normal menstrual cycles [13]. This study was underpowered to detect differences

**Table 4. Maternal and perinatal outcomes (Intention-to-treat analysis).**

| Clinical outcomes | NC | | HRT | | Absolute difference/ mean difference (95% CI)[a] | Risk ratio (95% CI)[a] |
|---|---|---|---|---|---|---|
| | N | N (%)/mean (SD) | N | N (%)/mean (SD) | | |
| Maternal hyperthyroidism,* | 248 | 6 (2.4) | 201 | 4 (2.0) | 0.4 (−2.3, 3.1) | 1.22 (0.35, 4.25) |
| Maternal hypothyroidism* | 248 | 31 (12.5) | 201 | 19 (9.5) | 3 (−2.7, 8.8) | 1.32 (0.77, 2.27) |
| Polyhydramnios* | 241 | 13 (5.4) | 195 | 9 (4.6) | 0.8 (−3.3, 4.9) | 1.17 (0.51, 2.68) |
| Oligohydramnios* | 241 | 5 (2.1) | 195 | 7 (3.6) | −1.5 (−4.7, 1.7) | 0.58 (0.19, 1.79) |
| Gestational diabetes mellitus | 248 | 27 (10.9) | 201 | 33 (16.4) | −5.5 (−12.0, 0.9) | 0.66 (0.41, 1.06) |
| Hypertensive disorders of pregnancy | 248 | 23 (9.3) | 201 | 16 (8.0) | 1.3 (−3.9, 6.5) | 1.17 (0.63, 2.14) |
| Pregnancy-induced hypertension | 248 | 15 (6.1) | 201 | 12 (6.0) | 0.1 (−4.3, 4.5) | 1.01 (0.49, 2.11) |
| Pre-eclampsia | 248 | 8 (3.2) | 201 | 4 (2.0) | 1.2 (−1.7, 4.2) | 1.62 (0.50, 5.31) |
| Antepartum hemorrhage | 242 | 35 (14.5) | 195 | 45 (23.1) | −8.6 (−16, −1.2) | 0.63 (0.42, 0.93) |
| Placenta previa | 242 | 2 (0.8) | 195 | 2 (1.0) | −0.2 (−2, 1.6) | 0.81 (0.11, 5.67) |
| Placenta accreta | 242 | 25 (10.3) | 195 | 29 (14.9) | −4.5 (−10.8, 1.8) | 0.69 (0.42, 1.15) |
| Unexplained | 242 | 8 (3.3) | 195 | 14 (7.2) | −3.9 (−8.1, 0.4) | 0.46 (0.20, 1.08) |
| Postpartum anemia* | 242 | 16 (6.6) | 195 | 16 (8.2) | −1.6 (−6.6, 3.4) | 0.81 (0.41, 1.57) |
| Preterm birth | 248 | 22 (8.9) | 201 | 25 (12.4) | −3.6 (−9.3, 2.2) | 0.71 (0.41, 1.23) |
| Spontaneous | 248 | 12 (4.8) | 201 | 15 (7.5) | −2.6 (−7.1, 1.9) | 0.65 (0.31, 1.35) |
| Medical reasons | 248 | 10 (4.0) | 201 | 10 (5.0) | −0.9 (−4.8, 2.9) | 0.81 (0.34, 1.91) |
| PPROM* | 241 | 12 (5.0) | 195 | 14 (7.2) | −2.2 (−6.7, 2.3) | 0.69 (0.33, 1.46) |
| Mode of delivery, cesarean section* | 242 | 171 (70.7) | 195 | 150 (76.9) | −6.3 (−14.5, 2.0) | 0.92 (0.82, 1.03) |
| Gestational age at birth (weeks) * | 242 | 38.86 (1.75) [b] | 195 | 38.72 (1.89) c | 0.14 (−0.20, 0.48) | – |
| Singleton | | | | | | |
| Birth weight (g) | 232 | 3354.7 (545.6) | 180 | 3344.7 (519.7) | 9.98 (−94.37, 114.34) | – |
| Low birth weight (<2500g) | 232 | 14 (6.0) | 180 | 13 (7.2) | −1.2 (−6.1, 3.7) | 0.84 (0.42, 1.73) |
| Very low birth weight (<1500g) | 232 | 1 (0.4) | 180 | 1 (0.6) | −0.1 (−1.5, 1.3) | 0.78 (0.05, 12.32) |
| High birth weight (>4000g) | 232 | 19 (8.2) | 180 | 13 (7.2) | 1 (−4.2, 6.1) | 1.13 (0.58, 2.23) |
| Very high birth weight (>4500g) | 232 | 3 (1.3) | 180 | 1 (0.6) | 0.7 (−1.1, 2.6) | 2.33 (0.24, 22.19) |
| Large for gestational age | 232 | 38 (16.4) | 180 | 27 (15.0) | 1.4 (−5.7, 8.4) | 1.09 (0.69, 1.72) |
| Small for gestational age | 232 | 10 (4.3) | 180 | 6 (3.3) | 1 (−2.7, 4.7) | 1.29 (0.48, 3.49) |
| Congenital anomaly | 232 | 7 (3.0) | 180 | 5 (2.8) | 0.2 (−3, 3.5) | 0.92 (0.30, 2.85) |
| Twins[b] | | | | | | |
| Birth weight (g) | 20 | 2500.5 (395.8) | 30 | 2438.7 (382.1) | 61.83 (−215.55, 339.21) | – |
| Low birth weight (<2500g) | 20 | 7 (35.0) | 30 | 14 (46.7) | −11.7 (−46.1, 22.8) | 0.75 (0.31, 1.80) |
| Very low birth weight (<1500g) | 20 | 0 (0.0) | 30 | 0 (0.0) | – | – |
| High birth weight (>4000g) | 20 | 0 (0.0) | 30 | 0 (0.0) | – | – |
| Very high birth weight (>4500g) | 20 | 0 (0.0) | 30 | 0 (0.0) | – | – |
| Large for gestational age | 20 | 2 (10.0) | 30 | 0 (0.0) | – | – |
| Small for gestational age | 20 | 5 (25.0) | 30 | 2 (6.7) | 18.3 (−4.6, 41.3) | 3.75 (0.78, 17.96) |
| Congenital anomaly | 20 | 4 (20.0) | 30 | 4 (13.3) | 6.7 (−24.1, 37.5) | 1.50 (0.24, 9.31) |
| NICU admission* | 241 | 31 (12.9) | 196 | 29 (14.8) | −1.9 (−8.5, 4.6) | 0.87 (0.54, 1.39) |
| Perinatal mortality | 0 | 0 (0.0) | 0 | 0 (0.0) | – | – |

[a]HRT group was regarded as the reference group.

[b]Generalized estimating equation for twins. Data represent the number of twin pairs.

*Post hoc specified endpoints.

Abbreviations: NC, natural cycle; HRT, hormone replacement treatment; CI, confidence interval; PPROM, preterm premature rupture of membranes; NICU, neonatal intensive care unit.

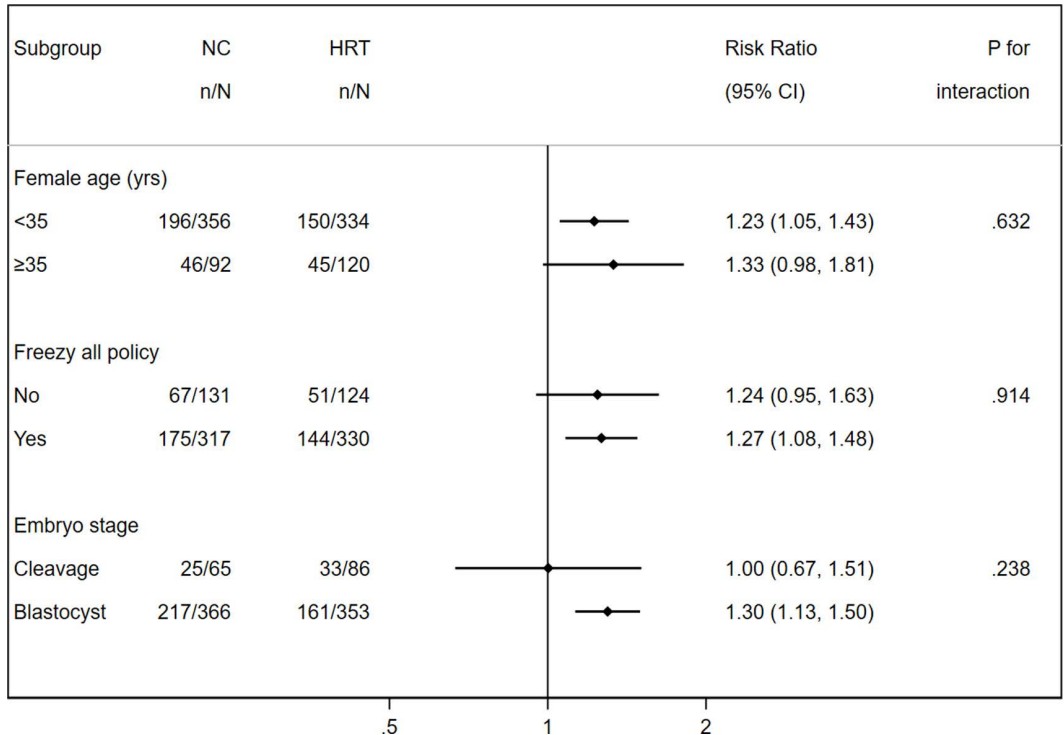

| Subgroup | NC<br>n/N | HRT<br>n/N | | Risk Ratio<br>(95% CI) | P for<br>interaction |
|---|---|---|---|---|---|
| **Female age (yrs)** | | | | | |
| <35 | 196/356 | 150/334 | | 1.23 (1.05, 1.43) | .632 |
| ≥35 | 46/92 | 45/120 | | 1.33 (0.98, 1.81) | |
| **Freezy all policy** | | | | | |
| No | 67/131 | 51/124 | | 1.24 (0.95, 1.63) | .914 |
| Yes | 175/317 | 144/330 | | 1.27 (1.08, 1.48) | |
| **Embryo stage** | | | | | |
| Cleavage | 25/65 | 33/86 | | 1.00 (0.67, 1.51) | .238 |
| Blastocyst | 217/366 | 161/353 | | 1.30 (1.13, 1.50) | |

**Fig 2. Forest plot of subgroup analyses on live birth rate (intention-to-treat).**

among four arms with a total of 471 patients. Furthermore, all transferred embryos were at the cleavage stage, and luteal phase support was provided through intramuscular progesterone administration, which is less commonly used in current practice in China. A retrospective cohort study investigating the efficacy of different protocols in eumenorrheic women reported that NC is superior to HRT in improving live birth rates [14], which is consistent with the findings in this trial. A recent large 3-arm RCT from Vietnam compared frozen embryo transfer in an NC, modified NC (with hCG to trigger ovulation), or HRT cycle and reported live birth rates of 37%, 33%, and 34%, respectively, with cancelation rates of 20% in the NC arms, leading to conversion to HRT [15]. While in that study lack of statistical significance resulted in a neutral interpretation, the 4% higher live birth rate and the 20% cancelation rate after in the NC strategy are in line with our conclusion that a strategy starting with NC endometrial preparation results in higher live birth rates than endometrial preparation with HRT. It is important to note that the population in our study is generally young, and over half had tubal infertility, which represents a more favorable prognosis following IVF. Tubal infertility, which includes conditions where the fallopian tubes are blocked or when the tubes are unable to collect an oocyte from the ovary due to adhesions in the pelvis, is responsible for 11% to 67% of cases diagnosed with infertility [16]. In China, a history of cervicitis, endometritis, previous stillbirths, and miscarriages has been significantly associated with an increased risk of infertility [17].

The results of our study support the routine use of NC instead of HRT for FET in ovulatory women. Our findings have significant implications for current clinical practice in IVF. Until now, physicians have the freedom to choose the endometrial preparation protocol for FET cycles based on their personal preference. The HRT protocol is often the preferred choice because of its flexibility in determining the timing of embryo transfer. However, our study provides evidence against this approach and suggests that NC should be chosen over HRT cycle.

We found that the miscarriage rate was significantly higher in women for whom the endometrium was prepared with HRT protocol than NC. It has been shown that high serum estradiol levels could damage the endometrium and the

window of implantation [18]. Estradiol levels in the HRT cycles were higher than in the NC cycles [19], and both estradiol and progesterone levels are aberrant in the HRT cycles [20,21]. Abnormal endometrial receptivity is proven to be a major contributor to miscarriage [22]. In addition, abnormal hormone levels can cause resistance of fetoplacental blood flow and decreases endometrial receptivity, which in turn, leads to miscarriage [23].

In our trial, antepartum hemorrhage in the HRT group occurred more frequently than in NC. The incidence of GDM and preterm birth were also numerically higher in the HRT group, though the differences were not statistically significant. Alternatively, it may be due to the absence of a significant difference in the risk of hypertensive disorders of pregnancy between the NC and HRT cycles for FET in ovulatory women, with population and selection biases minimized through the randomized controlled trial design. Further RCTs are necessary to clarify this important issue. Our results regarding adverse obstetric and perinatal outcomes correspond with other studies [24–26]. Previous studies have hypothesized that the detrimental effect of HRT on obstetric and perinatal outcomes could be mediated through suppression of ovulation leading to the absence of corpus luteum [27]. The absence of corpus luteum could have a deleterious effect on vascular health and renal adaptation during the first trimester [28,29]. Also, healthy placenta development requires the physiological change of estrogen and progesterone, the level of which is altered in HRT, and can cause issues with the placenta, including placenta accreta, placenta previa, and placenta abruption [30,31]. High estradiol may disrupt and early close the window of implantation [18], and have been shown to be associated with adverse pregnancy outcomes, including intrauterine growth restriction and pre-eclampsia [19]. Transcriptome analysis of the endometrium also suggested the superiority of NC over HRT in terms of gene expression for endometrial receptivity [32]. Another reason for the better clinical outcomes in NC versus HRT is perhaps because NC allows physiologically produced estradiol which is not subject to the fluctuations that are determined by patient compliance or the variable pharmacokinetics associated with orally administered estradiol in HRT [33]. All these could lead to suboptimal growth of endometrium and changes in the window of implantation.

There was no significant difference in hypertensive disorders of pregnancy, preterm birth, LGA, and SGA between the two groups. This result is different from previous studies [34]. The reason for the lack of significant differences may be due to the younger age of the patients we enrolled, with an average age of 30–31 years, inherently lowering the risk of complications during pregnancy. It remains possible that a difference in the risk of hypertensive disorders between NC and HRT exists in an older population or those who are anovulatory. Further RCTs are necessary to clarify this important issue.

The COMPETE trial has several strengths. One of the main advantages of this study is its adequate power, which enables it to provide precise estimates. Additionally, the important confounding factors were similarly distributed in both protocols, increasing the validity of the study. There is also likely minimal performance bias given the similar cycle characteristics between the protocols. COMPETE was conducted during the COVID-19 pandemic, which presented substantial challenges for completing the trial. Despite these challenges, there was a low cycle cancelation rate and high follow-up rate. Also, this study adopted a pragmatic approach, reflecting the routine IVF practice by allowing switching the endometrial preparation method given clinical conditions such as no ovulation rather than canceling the cycle.

The findings of our study were limited by the fact that all transfer cycles were performed at a single IVF center, which may affect the generalizability. The inclusion of both true NC and modified NC (with hCG trigger) reduces the power to draw conclusions regarding a specific type of NC FET. However, this approach ensures our study population closely mirrors the diverse clinical practices and patient scenarios encountered in real-world settings. The inclusion of such patients enhances the generalizability and applicability of our findings to a broader clinical audience.

Another limitation is that 149 women in total did not receive the assigned protocol. Most protocol violations occurred due to cross-over from NC to HRT, while fewer violations were observed during the transition to mild stimulation and GnRH agonist pretreatment before HRT. These permitted protocol violations, while potentially affecting the certainty in estimating the true efficacy of NC compared to HRT, closely align with real-world practice and prioritize patient interests. Also, the per-protocol analyses yielded similar results.

Crossovers happened because disturbances in the ovulation following superovulation are frequently observed in women with regular menstrual cycle [35]. In our study, among participants allocated to the NC group, 132 individuals (29.5%) underwent immediate transfer after oocyte retrieval. Similarly, in the HRT group, 122 participants (26.9%) underwent immediate transfer. These patients opted to proceed with FET immediately following oocyte retrieval, without allowing a full menstrual cycle for natural ovulation to resume. Unlike a recent RCT that canceled 20% of NC cycles primarily due to no ovulation [15], our approach provided these patients with the opportunity to early switch to an HRT protocol, thereby rescuing cycles from cancelation. Our results indicate that protocol adjustments during monitoring can be implemented in the early stages of endometrial preparation, optimizing reproductive outcomes while reducing unnecessary cycle cancelations in FET. However, we acknowledge that an early shift from NC to HRT in the absence of a leading follicle could potentially overlook women with late ovulation. Regarding the transition to mild stimulation, in clinical practice, when women undergoing HRT develop a dominant follicle with a low growth rate, clinicians often initiate ovulation induction—characterized as mild stimulation—to facilitate ovulation. Additionally, a small number of women switched to GnRH agonist pretreatment before HRT to delay embryo transfer, as the study period coincided with the COVID-19 pandemic.

A further limitation is that this study is open-label, therefore potential performance bias cannot be ruled out. We masked the embryologists and physicians who did the embryo transfer to the assignment of the treatment group to reduce performance bias. One could argue that further masking participants and physicians could have further reduced bias. However, we considered this would not be feasible given that transvaginal ultrasound and serum LH tests were performed more frequently in NC protocol, and participants could therefore speculate which protocol they were on. There was a noticeable trend toward an increase in blastocyst embryo transfers within the NC group, which may be attributed to the open design of this trial. However, this difference did not reach statistical significance. Last, our study has limited statistical power to detect associations with specific pregnancy complications. To accurately assess the relationship between endometrial preparation protocol in FET and pregnancy complications such as pre-eclampsia, a larger study population would be necessary.

In conclusion, the results of this large randomized controlled trial show that starting with NC for endometrial preparation and allowing to switch to HRT in case of no ovulation leads to better pregnancy outcomes than starting with HRT protocol and switching to NC if spontaneous ovulation occurs. Given the cross-over between the two arms, as well as the fact that the per-protocol analysis did not meet the prespecified sample size threshold, certainty in the efficacy of NC compared to HRT is lower.

## Supporting information

**S1 Table. Summary of adverse events.**
(DOCX)

**S2 Table. Reproductive outcomes (Per-Protocol Analysis).**
(DOCX)

**S3 Table. Maternal and Perinatal Outcomes (Per-Protocol Analysis).**
(DOCX)

**S4 Table. Sensitivity Analysis of Reproductive Outcomes by only including First Frozen-thawed Embryo Cycles (Intention-To-Treat).**
(DOCX)

**S5 Table. Sensitivity Analysis of Maternal and Perinatal Outcomes by only including First Frozen-thawed Embryo Cycles (Intention-To-Treat).\**
(DOCX)

**S6 Table. Sensitivity Analysis of Reproductive Outcomes following cycles with embryo transferred (Intention-To-Treat).**
(DOCX)

**S1 Text. COMPETE Trial Statistical Analysis Plan.**
(DOCX)

**S2 Text. CONSORT Checklist.**
(DOCX)

**S3 Text. Protocol.**
(DOCX)

**S1 Data. Raw data for the COMPETE trial.**
(XLSX)

## Acknowledgments

This study was overseen by an independent Data and Safety Monitoring Board (DSMB) and approved by the Northwest Women's and Children's Hospital. We gratefully thank Yuhua Shi (Guangdong Provincial People's Hospital, Guangzhou, China), Cuifang Hao (Reproduction Medical Center, Yantai Yuhuangding Hospital of Qingdao University, Yantai, China) and Yihong Guo (Center for Reproductive Medicine, Henan Key Laboratory of Reproduction and Genetics, The First Affiliated Hospital of Zhengzhou University, Zhengzhou, China) for their contribution as DSMB. We thank all the physicians, scientists, and embryologists in our IVF clinic for their assistance with data collection as well the patients for participating in this study.

## Author contributions

**Conceptualization:** Xitong Liu, Wentao Li, Juanzi Shi.

**Data curation:** Wentao Li.

**Formal analysis:** Xitong Liu, Xu Cao, Tianxing Liu.

**Funding acquisition:** Xitong Liu.

**Investigation:** Xitong Liu.

**Methodology:** Wentao Li, Pengfei Qu, Danmeng Liu.

**Project administration:** Xitong Liu, Wen Wen, Ting Wang, Tao Wang, Ting Sun, Na Zhang, Dan Pan, Jinlin Xie, Xiaojuan Liu, He Cai, Xiaofang Li, Zan Shi, Rui Wang, Na Lu, Haiyan Bai, Rong Pan, Li Tian, Bin Meng, Xin Mu, Hongran Jia, Hanying Zhou.

**Resources:** Xitong Liu.

**Supervision:** Wentao Li, Ben W. Mol, Juanzi Shi.

**Visualization:** Xitong Liu.

**Writing – original draft:** Xitong Liu.

**Writing – review & editing:** Xitong Liu, Wentao Li, Ben W. Mol.

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
