## [Editor Report · Decision Letter 0]

16 Jul 2024

Dear Dr Shi,

Thank you for submitting your manuscript entitled "Hormone replacement therapy versus natural cycle as endometrial preparation in women undergoing frozen-thawed embryo transfer: the COMPETE randomized controlled trial" for consideration by PLOS Medicine.

Your manuscript has now been evaluated by the PLOS Medicine editorial staff and we are would like to consider your submission further. Before we can make a final decision regarding peer review, we require additional information on when the statistical analysis plan was finalised and when trial data were available to the investigators. Please provide dates.

We also need you to complete your submission by providing the metadata that is required for full assessment. To this end, please login to Editorial Manager where you will find the paper in the 'Submissions Needing Revisions' folder on your homepage. Please click 'Revise Submission' from the Action Links and complete all additional questions in the submission questionnaire.

Please re-submit your manuscript within two working days, i.e. by Jul 18 2024 11:59PM.

Once your full submission is complete, your paper will undergo a series of checks.

Feel free to email me at lgaynor@plos.org if you have any queries relating to your submission.

Kind regards,

Louise Gaynor-Brook, MBBS PhD

Senior Editor

PLOS Medicine

---

## [Decision Letter · Decision Letter 1]

25 Oct 2024

Dear Dr Shi,

Many thanks for submitting your manuscript "Hormone replacement therapy versus natural cycle as endometrial preparation in women undergoing frozen-thawed embryo transfer: the COMPETE randomized controlled trial" (PMEDICINE-D-24-02242R1) to PLOS Medicine. The paper has been reviewed by subject experts and a statistician; their comments are included below and can also be accessed here: [LINK]

After discussing the paper with the editorial team, I'm pleased to invite you to revise the paper in response to the reviewers' comments. We plan to send the revised paper to some or all of the original reviewers, and we cannot provide any guarantees at this stage regarding publication.

We ask that you submit your revision by Nov 15 2024 11:59PM. However, if this deadline is not feasible, please contact me by email, and we can discuss a suitable alternative.

Don't hesitate to contact me directly with any questions (lgaynor@plos.org).

Best regards,

Louise

Louise Gaynor-Brook, MBBS PhD

Senior Editor

PLOS Medicine

lgaynor@plos.org

Comments from the editors:

As you will see below, two of the reviewers have questioned the validity and plausibility of the data and conclusions presented in this manuscript. Please ensure to address the comments raised by the reviewers in full prior to resubmission. As part of the resubmission, we request that you provide a completely anonymised dataset in a format that can be shared with the reviewers and published alongside the manuscript, should we accept the manuscript for publication (please note that we do not provide any guarantees regarding publication at this stage). We were concerned that the dataset provided by email may provide potentially identifiable information, and we request revision of this dataset to ensure complete anonymity for participants in the trial, to allow reviewers to maintain their anonymity (i.e. that they are not required to request access to a data repository), and to ensure complete compliance with the ethical approvals of the trial. This request for the data to be shared with the reviewers is a requirement for further consideration at the journal.

We note also that there are six co-first authors. Please ensure that all co-authors have declared their contributions under the usual ICMJE guidelines, and please verify that the contribution of the six co-first authors were equal.

We note that some outcomes presented were not pre-specified = maternal and perinatal outcomes including maternal thyroid status, polyhydramnios, oligohydramnios, PPROM, mode of delivery, gestational age at birth, NICU admission, postpartum anaemia. These are pre-specified only in the SAP as post hoc analyses; please ensure that these are presented as post-hoc analyses in the manuscript. Perinatal mortality was a pre-specified outcome but not included in the manuscript - please clarify why this outcome has been omitted.

The SAP appears to have been finalised on 10 Sept 2023 (the last follow-up in the trial was Oct 2023); please indicate in the manuscript that the SAP was finalised prior to data lock / independently of the data.

Comments from the reviewers:

Reviewer #1: The manuscript describes the results for the parallel-arm, open-label, randomised controlled trial COMPTE, evaluating 2 methods of endometrial preparation for women undergoing froze-thawed embryo transfer.

This is a statistical review and hence the focus is primarily on methodological and reporting aspects of the manuscript.

OVERALL ASSESSMENT

This is a generally well run trial and well described; I did not find any real dealbreakers during my review.

The CONSORT checklist compliance needs to be addressed as does the need for a bit more clarity on statistical methods (taken together these required revisions probably clear the bar for major revision, but definitely on the lower end as far as major revisions are concerned).

REGISTRATION

I confirm that I checked the trial registration with the Chinese Clinical Trial Registry (CHiCTR2000040640) - this is OK.

PRE-SPECIFICATION: PROTOCOL & STATISTICAL ANALYSIS PLAN (SAP)

I compared the results from the manuscript with what was written in the protocol - the primary and secondary outcomes are the same, with a few exceptions for secondary outcomes but all of these are transparently acknowledged in the manuscript.

The SAP was only provided as a supplementary document and was not deposited on the trial registry as far as I was able to assess. the authors state this was finalised on 17 September 2023 before database lock, but there is no independent way to verify this. That said, the SAP largely recapitulates, with a few more details, what was in the protocol. The SAP does highlight the additional post-hoc outcomes (all transparently highlighted by the authors in the main manuscript) and states a reason why they were included.

CONSORT CHECKLIST

Below I go through those items on the CONSORT checklist which are currently not adequately covered by the manuscript. These are mostly minor clarifications and not major issues.

Item 1b: The required info is in the abstract but not 100% compliant with the CONSORT required structure. Obviously there are PLOS Medicine abstract structures to comply with as well, so I leave this one for the PLOS editors to decide on.

Item 2b: The background section currently does not explicitly define objectives and hypotheses though is very clear on the rationale for the study.

Item 3a: All the required info (parallel arm, 1:1 randomised, unblinded controlled trial) is given in the manuscript, but CONSORT requires it to be stated in the trial design section, which is currently not the case. E.g. some of this info is currently in the randomisation section.

Item 4b: Again all the info is provided but not in the section required by CONSORT. Currently the Participants section does not identify the setting & location where participants were recruited from.

Item 6b: The authors transparently highlight post-hoc outcomes on the results tables. However, 1) this info should also be stated in the outcomes section, together with a reason for including the post-hoc outcomes (currently this reason is only given in the supplementary SAP); 2) perinatal mortality is specified in the SAP and the manuscript does state that there were no such cases, but probably Table 4 should still explicitly list this secondary outcomes as it was pre-specified.

Item 7a: The sample size calculation information is included at the start of the Statistical Analysis section but really should be a section on it own.

Item 8a: This is included but could be more specific (currently only states what software was used for this but no further technical details).

Item 10: Currently the information on who generated the random allocation sequence, who enrolled participants and who assigned participants to interventions is missing.

Item 12a: Statistical methods are explained in the manuscript but are not 100% clear. It is currently not clear

1) when a chi-squared test and when a Fisher's exact test was used (presumably based on expected counts exceeding some threshold for the chi-square test but this is not specified anywhere; it would also be good if Tables 3 & 4 could explicitly highlight which test was used for which outcome),

2) why robust Poisson regression models were used for binary outcomes for adjusted analysis (the reader has to infer that this was probably because these are less sensitive to outliers but would be good to have this spelled out),

3) when the regression models were used,

4) what the exact details of the generalised estimating equations (which are an estimation method rather than a model) that were used for the analysis of twins data were,

5) how the interaction p-values were computed in the sub-group analyses; this is currently quite unclear.

Item 14b: Technically this is missing, though it is obviously very clear that the trial ended when the sample size was reached (though technically for both groups the sample size of 444 per group was slightly exceeded - reasons for this should be stated).

STATISTICAL DESIGN & ANALYSIS

Overall, the design and analysis for this trial are appropriate and explained in the manuscript. Details of the statistical analyses used should be clearer (see comment above for Item 12a for the CONSORT checklist).

I found 2 minor issues with the sample size calculation:

1) I did get a marginally different sample size estimate (375 per group rather than 370) but this could be due to having used a different sample size formula than the one the authors used,

2) The adjustment for 20% LTFU is done incorrectly. The authors used 370 and multiplied this by 1.2 (which gives 444), but they should have divided by 0.8 (which would have given 463). LTFU in the end was 0% - not a single participant dropped out (is that right?) - and so this point is anyway a bit moot.

Statistical significance is only defined implicitly (through the use of 95% CIs) as 5%, so would be preferable to state this explicitly somewhere in Methods.

I am fine without p-values as this is clear from the CIs, but I wonder if adding a p-value column to Tables 3 and 4 would be clearer rather than just using bold font to highlight statistically significant results. that would allow showing an exact p-value rather than just <0.05 or >=0.05.

On Tables 3 & 4, the authors mix the use of proportions and percentages (e.g. the columns by study arm state percentages, but the absolute difference is given as a difference in proportions) which is confusing, and I would recommend using consistently proportions (or consistently percentages).

There seems to be a mistake on the last row ("Multiple pregnancy") of Table 3: The percentage in the NC group should be 4.3% (19/448) and consequently the absolute difference and risk ratio are in the wrong direction (these seem to be for the per protocol analysis?).

FURTHER COMMENTS

1. Neither the underlying data nor data analysis code are made publicly available. Data can be available "upon reasonable request" but no information provided how to obtain the analysis code that was used by the authors.

2. Having 6 co-first authors is quite extraordinary. Further, this is not supported by the author contribution statement which lists many more contributions for the first 2 of the 6 co-first authors. This seems unfair to these 2 authors. I am aware that author contribution statements cannot fully capture exactly what every author contributed, but it looks as if the statement conflicts with the co-first status.

3. The unit of randomisation and analysis are the women undergoing the FET treatment. However, there seems to be an implicit assumption that every women had a male partner and so the text and results tables make no difference between women, couples and male partners. If it is the case that every woman included in the study had a male partner, then this should be stated explicitly somewhere as otherwise it can be a bit confusing to the reader (who might assume that some women would have undergone the treatment using donated sperm).

4. On p.14, where the authors describe using generalised linear models, there are 2 typos: 1) it should be "adjusted" not "unadjusted" risk ratio presumably as the unadjusted analyses used simpler tests -- also there is a need to specify what variables were adjusted for, 2) "Poisson log link" should be replaced with "Poisson distribution and log link function".

5. I wonder if a post-hoc sensitivity analysis excluding the women who did not perform embryo transfer should be completed and included in the supplementary for the sake of completeness. Clearly will not change results but seems an obvious thing to do.

6. Figure 2 is not being referred to in the main text. In fact, only the Discussion section refers to these sub-group results. the results should be stated in the Results section and refer to Figure 2 and the Discussion section can then pick this up again.

7. On p.20 where the statistically non-significant results for diabetes mellitus and preterm birth are referred to, this feels a bit like cherry-picking as these two had directions that agreed with the narrative of the paper, whereas the result for hypertensive disorders, which is also not statistically significant, has a less convenient direction of effect. So I would either remove referring to these results at all or, preferably, mentioning the hypertensive result as well.

(VERY) MINOR COMMENTS

1. On p.13 it says "If a participant did not reply after multiple contact attempts [...]". Please specify how many.

2. On p.15, where the mean ages (30.7 and 31.3) are given, the text should state which mean is for which group. Currently the reader has to assume the first is for the NC group and the second for the HRT group or verify with Table 1.

3. On p.17 where it says "were not significantly different", I would be very clear that this is statistical significance: "were not statistically significantly different".

4. On p. 21 where it says "confounding factors were equally distributed" I would write "confounding factors were similarly distributed".

Reviewer #2: RCT of the two main FET protocols showing benefit of NC.

Obviously recent similar RCT from Vietnam, but still useful data esp as both underpowered for many pregnancy complication outcomes so useful for future metanalyses.

"or to endometrial stimulation induced by HRT " Is stimulation the correct phrasing?

"Women with ovulation disorders, defined as irregular ovulation, anovulation, or the absence of ovulation, were not eligible, as were women with intrauterine adhesions diagnosed at hysterosalpingography, hysteroscopy, or transvaginal ultrasound." Were these routinely assessed? If so how?

"ovulation + 3 day, LH surge + 4 day," Please explain how you define ovulation here?

please add birth weight definitions to methods as per the table.

Can you mention in limitation that to properly power to pregnancy complications eg pre-eclampsia for example would require a larger study.

Table 1 How was diminished ovarian reserve defined?

What was done about abnormal TSH?

What does other protocol refer to?

Reviewer #3: Dear Sirs

I congratulate the authors for their huge contribution to science with this very important RCT.

However, I find it very important that the conclusions on obstetric and neonatal outcomes stays in line with the findings of the RCT as RCTs has such a strength as compared to retrospective studies with regards to selection/population bias that the truth is most likely found in RCTs rather than in retrospective cohort studies.

In the presented RCT, especially HDP including pre-eclampsia, and risk of preterm birth, LGA and SGA do NOT differ between groups in ovulatory, young (inclusion criteria <35 years) women, and this is very interesting and should be clearly passed on to the readers as this is quite different from results of recent meta-analyses (Zaat et al. 2023) but may in fact be closer to the truth (although more RCTs will be important before final conclusions). There was only a significant difference in antepartum haemorrhage (and a non-significant difference in gestational diabetes).

The finding of improved LBR with NC FET seems to be far enough reason to perform NC FET.

'Searching for the truth', the data presented in the present manuscript/tables would benefit for a more detailed presentation of data regarding the following issues - and if relevant adjusted analyses could be performed:

The following should be addressed in the paper:

How many were transferred with day 5/6/7 vitrified blastocysts in each group?

How many had PGT-A (euploid embryo transfers (blastocysts, I assume)) in the two groups?

How many had FET in the cycle immediately following ovarian stimulation and oocyte retrieval in both groups?

How many had NC based on LH surge and ultrasonic follicle collapse and how many had mNC in the NC group?

How many in the HRT group had estradiol dose increment to 8 mg per day? (and was estradiol stopped in the few cases with follicular growth in the HRT group?)

Abstact

Aim: we aim to assess whether reproductive?, obstetric and perinatal outcomes differ between natural cycles (NC) and hormone replacement cycles (HRT) in women with a regular ovulatory cycle

Author summery:

Stating that HRT increases obstetric and perinatal complications compared with NC based on the presented data seems not valid.

It should be clear from the manuscript itself,

* if all embryos were vitrified and warmed the same day as the transfer

* if women were randomized only once

Other:

Was hyper- hypothyroism newly diagnosed or were women in relevant medical treatment?

"The primary outcome was live birth resulting from the first FET" - were women included more than once?

"Prespecified exploratory subgroup analyses included age group (<35 years vs ≥ 35 years)" - women were only included if <35 years?

"IVF centers may benefit from discontinuing the standard use of HRT for FET. The benefit would include the avoidance of extra medical expenses as well as the burden of obstetrical and perinatal complications." The costs may not differ a lot as luteal phase progesterone was administered in NCs. My comments about obstetrical and perinatal complications above. I think the discussion should be more balanced including your own findings.

Definition of ovulatory cycle, and comment on the high proportion of women not ovulating

Definition of tubal factor, and comment on the high proportion of tubal factor

Reviewer #4: In this open label RCT, in women with a regular menstrual cycle undergoing IVF, the live birth rate following frozen embryo transfer (FET) in and natural cycle was compared with the live birth rate following FET in a programmed HRT cycle. The intention to treat analysis indicated that natural cycle preparation for FET was associated with 54% live birth rate whereas transfer following HRT preparation resulted in a significantly lower chance of achieving a live birth rate at 43% in the HRT group.

The rationale for performing such a trial is understandable given most literature on the subject have not addressed the question whether a particular method of preparation for FET is superior to the other. However, the findings of this raise at least two pertinent questions a question. The first one is about biological plausibility of achieving this remarkably high live birth rate following FET whether it is following natural cycle or after HRT preparation. The live births rates reported here following frozen FET have not be reported by any previous study on the subject to date. Moreover, the significant difference between the natural and HRT prepared FET has not previously been reported in any other RCT. In fact, a recently published, adequately powered study, no significant difference in the livebirth outcome between natural and HRT prepared FET was found. The livebirth rate after one FET was 174 (37%) of 476 in the natural cycle strategy group, 159 (33%) of 476 in the modified natural cycle strategy group, and 162 (34%) of 476 in the artificial cycle strategy group (1).

It is interesting such high live birth rate and the significant difference in outcome of FET in favour of natural cycle FET were only in addition to this manuscript, was reported in published cohort and retrospective studies from the centre where this current study has been conducted (2)

The above observations question the biological plausibility of the result and raises a question regarding potential intellectual bias, both of which make a case for examining the raw data which I hereby renew my request to make available in order to complete an objective assessment of the data underpinning the findings of this study.

1-Vu N A Ho, et.al., Lancet 2024; 404: 266-75

2-liu X, Sh W, SheJ Fertil Steril. 2020:113 (4):811-7

Reviewer #5: This is the report of a randomized trial of natural cycle vs programmed cycle for FET. The trial does have some methodologic issues which raise some questions about the validity of the conclusions.

Methods

1. Patients should all have been ovulatory; why did about 1/4 of the patients randomized to the NC arm not ovulate and get switched to the HRT arm?

2. The NC arm in fact included various regimens including hcg, or natural ovulation, or the use of cleavage stage embryos or blastocysts- leading to more variables than just a natural ovulation

3. The number of embryos transferred varied

4. It is unclear which group the 105 NC patient who did not ovulate were included in- please clarify

Results

5. " significant reduction in antepartum haemorrhage in the NC group compared to the HRT group" There is not a reduction- reduction means that something within a group changed from a prior measurement. What you mean is that the NC group had a lower incidence of hemorrhage.

6. It appears that the NC group had some what better embryo quality than the HRT group based on the table, with more blastocyst transfers and more single embryo transfers- though differences were NS at 0.079

Discussion

7. The discussion must discuss problems with the paper including the lack of a unified NC protocol,

8. The discussion must discuss and how the 105 natural cycle patients converted to HRT were analyzed and included in the results, or excluded, and how this might impact the results

* Please upload any figures associated with your paper as individual TIF or EPS files with 300dpi resolution at resubmission; please read our figure guidelines for more information on our requirements: http://journals.plos.org/plosmedicine/s/figures. While revising your submission, please upload your figure files to the PACE digital diagnostic tool, https://pacev2.apexcovantage.com/. PACE helps ensure that figures meet PLOS requirements. To use PACE, you must first register as a user. Then, login and navigate to the UPLOAD tab, where you will find detailed instructions on how to use the tool. If you encounter any issues or have any questions when using PACE, please email us at PLOSMedicine@plos.org.

* PLOS Medicine requires that the de-identified data underlying the specific results in a published article be made available, without restrictions on access, in a public repository or as Supporting Information at the time of article publication, provided it is legal and ethical to do so. Please revise your Data Availability Statement.

FIGURES AND TABLES

SUPPLEMENTARY MATERIAL

REFERENCES

RCTs

* PLOS Medicine requires that all trials be prospectively registered in one of registries recognized by WHO. Please ensure that study registration details are included in the Methods section.

* Please structure the Methods section using the following sub-headings: Study design and participants, Randomization and masking, Procedures, Outcomes, Statistical analysis.

* Please clarify and explain all discrepancies between the paper and protocol. If the outcomes were not prespecified in the protocol, please define them in the Methods (Outcomes section) as post hoc and explain why they were added. Post-hoc comparisons should be presented as hypothesis generating rather than conclusive.

* Please ensure that all prespecified outcomes (primary, secondary, and exploratory) are listed in the Methods/Outcomes section and indicate whether there are outcomes that are not presented in the current report.

* Please specify the dates (Month Day, Year) during which study enrolment and follow up occurred.

* Please include absolute numbers wherever you report percentages; eg, n/N (%)

* Please present the safety data for the study including numbers of specific events and whether or not adverse events are thought to be related to treatment. AEs should be reported in the abstract, per CONSORT and CONSORT-Harms.

* Please complete the CONSORT checklist (https://www.equator-network.org/reporting-guidelines/consort/) and ensure that all components of CONSORT are present in the manuscript, including how randomization was performed, allocation concealment, blinding of intervention, definition of lost to follow-up, power statement. When completing the checklist, please use section and paragraph numbers, rather than page numbers.

* Please report your abstract according to CONSORT for abstracts, following the PLOS Medicine abstract structure (Background, Methods and Findings, Conclusions) https://www.equator-network.org/reporting-guidelines/consort-abstracts/

* If your trial had to undergo important modifications in response to extenuating circumstances, please complete the CONSERVE-CONSORT checklist and provide in your Supporting Information; (https://www.equator-network.org/reporting-guidelines/guidelines-for-reporting-trial-protocols-and-completed-trials-modified-due-to-the-covid-19-pandemic-and-other-extenuating-circumstances-the-conserve-2021-statement/). When completing the checklist, please use section and paragraph numbers, rather than page numbers.

* In keeping with our commitment to Open Science, please include the study protocol document and analysis plan (including any amendments) as Supporting Information to be published with the manuscript if accepted.

* Please note that PLOS Medicine requires prospective, public registration of a data sharing plan (as part of mandatory clinical trials registration) for all clinical trials that began enrollment on or after January 1, 2019, in accordance with ICMJE requirements.

---

## [Decision Letter · Decision Letter 2]

23 Jan 2025

Dear Dr Shi,

I am writing in place of my colleague Dr. Gaynor-Brook, who is away from the office. Thank you for submitting your revised manuscript "Hormone replacement therapy versus natural cycle as endometrial preparation in women undergoing frozen-thawed embryo transfer: the COMPETE randomized controlled trial" (PMEDICINE-D-24-02242R2) to PLOS Medicine. The paper has been reviewed by 3 subject experts and a statistician; their comments are included below and can also be accessed here: [LINK]

As you will see, the reviewers continue to cite concerns that need to be addressed for further consideration of the study. After discussing the paper with the editorial team, we invite you to revise the paper in response to the reviewers' and editors' comments. We plan to send the revised paper to some or all of the original reviewers, and we cannot provide any guarantees at this stage regarding publication.

Please note the following editorial concerns:

1. The code must be deposited in a public repository, such as GitHub. Please include the accession url in a revised manuscript.

2. The participant study group cross-overs must be framed as a limitation in the Abstract. Please also state in the Abstract that this is a single center study.

3. The conclusions in the Abstract and Discussion must be tempered in view of the high number of cross-overs. Please also note that the PP analysis results in groups failing to meet the prespecified sample size threshold.

4. Please clarify in the Methods whether preimplantation genetic testing was performed, and for how many women.

5. Please include an acknowledgment in the Introduction that the study is restricted to married women due to regulations in China.

6. Please provide some context on IVF and ART, and pronatalist policies, in China for the general reader, and how the latter might affect age and frequency of access to ART, and potential effects on LBR.

7. Please state directly in the Results that no age exclusion was applied to participant recruitment or enrollment.

8. The large degree of variability between the NC and HRT groups, not solely limited to the cross-over events, must be transparently discussed.

9. The numbers in each group in the PP analysis need to be more transparently explained to the general reader.

10. Please add open label to the title.

**11. We ask that you perform a post hoc instrumental variable reanalysis for the PP analysis (ITT random assignment as the instrument and exposure as treatment; report s statistic) to account for cross-over between the study arms. Please state in the text that this analysis was not prespecified and is post hoc.

Please also be advised that we will refer the issue of 6 potential first co-authors for further internal discussion.

We ask that you submit your revision by Feb 13 2025 11:59PM. However, if this deadline is not feasible, please contact me by email, and we can discuss a suitable alternative.

Don't hesitate to contact me directly with any questions (afarrell@plos.org).

Best regards,

Alison

Alison Farrell, PhD

Senior Editor

PLOS Medicine

afarrell@plos.org

Comments from the reviewers:

Reviewer #1: I thank the authors for revising and resubmitting their manuscript as well as sharing the underlying data. I am overall satisfied with their revisions to the points I raised (with 4 somewhat minor exceptions listed below) and I am happy with the manuscript from a statistical review point of view. I do note that Reviewer 4 has raised a more general, subject matter-specific potential issue and flags this study as potentially a surprising outlier among similar studies. My OK from a statistical point of view should of course not overrule any subject-matter specific concerns.

1. Regarding the details of the main analysis, I now understand that the authors used simple tests (Fisher's exact test or Chi-squred) to compute statistical significance only and then used the Poisson regression models for RR estimates and CIs for these estimates. The authors rightly state that due to the use of randomisation, adjustment is not strictly necessary (though note that it still may be if there are substantial random imbalances in covariate distributions as a result of the randomisation), but it would be more common to use simple standard formulas for the RRs and their CIs in this case rather than a regression model-based approach (which makes further, distributional assumptions; though for the binary data this approach was used for, this is admittedly not very relevant). Still, while I appreciate using the regression models, it would be more consistent to then report statistical significance based on these models rather than from the simpler tests - there is the potential, for more borderline associations, that the simpler test p-values conflict with the CIs from the regression models.

2. Regarding the sample size calculation, while I thank the authors for moving this to another section and clarifying on the differences between their and my calculations, I do think that the 10.1% absolute difference that was powered for should be mentioned rather than the current 10%. The power is for detecting a difference at least as large as the one powered for, and hence the exact value used should be stated (the 0.1% that the authors added is arbitrary and unclear -- e.g. why not adding 0.05% or 0.01% or 0.000001%?). Also I would make it clear that the 10.1% is an absolute rather than a relative difference in percentages.

3. Regarding my point about statistical significance and exact p-values, this point was made precisely because of the recommendation (e.g. https://doi.org/10.1080/00031305.2019.1583913) for a more nuanced analysis rather than a dichotomisation of results according to p<0.05 or p>=0.05. Through the bold emphasis currently employed by the authors in the results tables, I feel that this dichotomisation of results is encouraged, and hence why I suggested including exact p-values instead. I can understand this would overload the tables. But to follow the authors' response, which suggests they share a desire for a more nuanced analysis, I would then suggest to drop the bold emphasis (and hence p-values) altogether and let simply the RRs and their CIs speak for themselves.

4. I still think that having 6 co-first authors is exceptional but I leave this point to the editors.

Reviewer #2: no further comments

Reviewer #3: Dear Authors

Again, I congratulate you for your effort and great work. I think the manuscript has been improved. My last comments you will find below:

I propose an alternative title i) clarifying that only women with ovulatory cycles were included, and ii) writing the natural cycle first (for consistency with the text of the manuscript):

Natural cycle versus hormone replacement therapy as endometrial preparation in ovulatory women undergoing frozen-thawed embryo transfer: the 3 COMPETE randomized controlled trial

(Similarly in the Full Title)

Comments

The following phrases I find too unspecific:

L95-96: ….and decreased the risk of some obstetric and perinatal complications, com-pared to hormone replacement therapy cycle. (The miscarriage rate can be considered as an reproductive outcome, and should antepartum haemorrhage be considered as a complica-tion? It depends on the definition, in my opinion)

L 225-226: Blastocysts graded above 4CC were considered usable (only one C allowed in either the ICM OR TE score?)

L253: The primary outcome measure was live birth resulting from the first FET after inclu-sion?

Outcome definitions

As the only obstetric outcome significantly differing between groups is antepartum haemor-rhage, I find it important that the authors clearly define the outcome in the method section of the manuscript. (L261: ….., antepartum haemorrhage,….)

Further, it should be discussed in the manuscript if patient reported spotting or very small bleeding is included as 'yes' or 'no'. and if 'yes', it should be discussed whether the outcome should be considered as an obstric complication or not, i.e., in L460: In our trial, antepartum haemorrhage in the HRT group occurred more frequently 460 than in NC, followed by a rele-vant discussion of the impact of this important statement. Where there any more miscarriag-es, preterm deliveries or other complications among women with antenatal haemorrhage?

L480-484:….., or simply because there is no difference in risk of HDP between NC and HRT FET strategy in a population of ovulatory women, where population/selection bias is mini-mised by the randomised controlled design. Further RCTs are warranted to elucidate this im-portant question.

L499-500: I think you should stick to the relevant limitations here, i.e., The inclusion of both tNC and mNC FET cycles reduces the power to conclude on one specific type of NC FET, however, the results may be relevant to a broader clinical audience.

L506-507: you could add that the per-protocol analyses reached similar results

L516-519: I think the following sentence should be omitted as it cannot be concluded based on the literature references and it do not add any benefit/clarification to the paper: "This find-ing underscores the detrimental effect of ovarian stimulation on the normal ovulation cycle (e.g., ovarian cysts) and hypothalamic-pituitary-ovarian (HPO) axis function, as proven by previous studies [35, 36]. - Ref. 35 document a prolonged follicular phase in immediate vs. postponed FET (with similar reproductive outcomes thus not detrimental!), while in Ref. 36, the FET immediate group (embryo transfer <60 days after oocyte retrieval) may well include women with on natural cycle in between the fresh and the FET cycle). Instead of 'overinter-pretation', I recommend deletion of the sentence, and instead include a sentence where you mention that with a NC FET strategy like yours, with an early switch from NC to HRT FET if no leading follicle on cycle day 10, quite a few NC with late ovulation where most likely missed at the expense of early conversion to HRT (to minimize the risk of cancellation due to a sporadic anovulatory cycle)

Minor

L123 …risk of perinatal and neonatal complications - is it rather obstetric and neonatal com-plications?

L261: hemorrhage => haemorrhage

L364-365: For reproductive outcomes, participants in the NC had significantly thicker endo-metrium than in the HRT group. I would consider endometrial thickness as a cycle character-istic rather than a reproductive outcome!

L243-246 (if I understand the text correct, this section refers to both FET groups, if so you could move the section up just before L 206): In both FET groups, Dydrogesterone, at a dosage of 10 mg, was administered orally three times daily commencing from the day of em-bryo transfer. In the HRT group, oral estradiol was gradually diminished if clinical pregnancy was confirmed. Luteal phase support was continued until the 10th week of pregnancy.

L373: Regarding maternal and perinatal outcomes => Regarding obstetric and perinatal out-comes

L373: there was a significantly lower chance => there was a significantly lower risk

L 390 and L 394: maternal => obstetric

L401: that live birth rates were => that the live birth rate was

L414: frequent monitoring of follicular growth or LH surge => monitoring of follicular growth or LH surge

L451-452 We found that the miscarriage rate after HRT was significantly higher in whom the endometrium was prepared with HRT protocol than NC => We found that the miscarriage rate was

Reviewer #5: The authors have done a good job responding to most of the comments. Questions still remain as detailed below:

Reviewer 1

comment 7

1. How were patients randomized on CD 7? This may well be too late for a patient to receive their HRT and start it in time to ensure that they did not ovulate; what cycle day was HRT begun?

2. How did you ensure that patients on HRT did not have a spontaneous ovulation?

Comment 22

3. In my opinion this should say "incidence" of preterm birth not "risk" of preterm birth

Reviewer 2

comment 6

4. Advanced maternal age is not equal to decreased ovarian reserve, and is not accepted as part of the definition of ovarian reserve in our field. The authors point out that DOR is defined by elevated, FSH, low AMH, low antral follicle count. Women of 40 may have excellent ovarian reserve with high numbers of eggs retrieved. It is of course known that aneuploidy rates in eggs of advanced age women are higher, but this does not change their ovarian reserve. This needs to be changed where relevant in the study.

Reviewer 5

comment 1

5. Please clarify exactly when after an egg retrieval a NC or HRT cycle was started? If you mean with the menstrual period following a negative pregnancy test, please clearly state this.

Comment 2

6. Please clarify when cleavage stage transfers were performed. In the response to one of the other reviewers you stated that you performed only day 5 and 6 blastocyst transfer.

Comment 4

7. The fact that NC cycles were included in the ITT analysis needs to be clearly stated in the paper, not just referred to by implying that readers need to do the math when they read table 3 to determine that they were included.

---

* Please upload any figures associated with your paper as individual TIF or EPS files with 300dpi resolution at resubmission; please read our figure guidelines for more information on our requirements: http://journals.plos.org/plosmedicine/s/figures. While revising your submission, please upload your figure files to the PACE digital diagnostic tool, https://pacev2.apexcovantage.com/. PACE helps ensure that figures meet PLOS requirements. To use PACE, you must first register as a user. Then, login and navigate to the UPLOAD tab, where you will find detailed instructions on how to use the tool. If you encounter any issues or have any questions when using PACE, please email us at PLOSMedicine@plos.org.

* [EDITOR: CHECK FINANCIAL DISCLOSURES, COI, DAS, AND ETHICS STATEMENTS AND INCLUDE ANY NECESSARY REQUESTS]

* Please ensure that the study is reported according to the [XXXX] guideline and include the completed [XXXX] checklist as Supporting Information. When completing the checklist, please use section and paragraph numbers, rather than page numbers. Please add the following statement, or similar, to the Methods: "This study is reported as per [XXXX] guideline (S1 Checklist)."

FIGURES AND TABLES

SUPPLEMENTARY MATERIAL

REFERENCES

[STUDY TYPE-SPECIFIC REQUESTS - DELETE SECTIONS AS NECESSARY]

RCTs [REFER TO RCT CHECKLIST AND MEETING NOTES FOR DETAILS TO ADD]

* PLOS Medicine requires that all trials be prospectively registered in one of registries recognized by WHO. Please ensure that study registration details are included in the Methods section.

* Please structure the Methods section using the following sub-headings: Study design and participants, Randomization and masking, Procedures, Outcomes, Statistical analysis.

* The following outcomes measures [ADD DETAILS AS NEEDED OR DELETE BULLET POINT] appear to differ between the submitted manuscript and the protocol [and/or trial registry]. Please clarify and explain all discrepancies between the paper and protocol. If the outcomes were not prespecified in the protocol, please define them in the Methods (Outcomes section) as post hoc and explain why they were added. Post-hoc comparisons should be presented as hypothesis generating rather than conclusive.

* Please ensure that all prespecified outcomes (primary, secondary, and exploratory) are listed in the Methods/Outcomes section and indicate whether there are outcomes that are not presented in the current report.

* Please specify the dates (Month Day, Year) during which study enrollment and follow up occurred.

* Please include absolute numbers wherever you report percentages; eg, n/N (%)

* Please present the safety data for the study including numbers of specific events and whether or not adverse events are thought to be related to treatment. AEs should be reported in the abstract, per CONSORT and CONSORT-Harms.

* Please complete the CONSORT checklist (https://www.equator-network.org/reporting-guidelines/consort/) and ensure that all components of CONSORT are present in the manuscript, including how randomization was performed, allocation concealment, blinding of intervention, definition of lost to follow-up, power statement. When completing the checklist, please use section and paragraph numbers, rather than page numbers.

* Please report your abstract according to CONSORT for abstracts, following the PLOS Medicine abstract structure (Background, Methods and Findings, Conclusions) https://www.equator-network.org/reporting-guidelines/consort-abstracts/

* If your trial had to undergo important modifications in response to extenuating circumstances, please complete the CONSERVE-CONSORT checklist and provide in your Supporting Information; (https://www.equator-network.org/reporting-guidelines/guidelines-for-reporting-trial-protocols-and-completed-trials-modified-due-to-the-covid-19-pandemic-and-other-extenuating-circumstances-the-conserve-2021-statement/). When completing the checklist, please use section and paragraph numbers, rather than page numbers.

* In keeping with our commitment to Open Science, please include the study protocol document and analysis plan (including any amendments) as Supporting Information to be published with the manuscript if accepted.

* Please note that PLOS Medicine requires prospective, public registration of a data sharing plan (as part of mandatory clinical trials registration) for all clinical trials that began enrollment on or after January 1, 2019, in accordance with ICMJE requirements.

OBSERVATIONAL STUDIES

* Abstract: Please include the study design, population and setting, number of participants, years during which the study took place (enrollment and follow up), length of follow up, and main outcome measures.

* Please ensure that the study is reported according to the STROBE (or appropriate STOBE extension) guideline (available from: https://www.equator-network.org/reporting-guidelines/strobe) and include the completed STROBE (or STROBE extension) checklist as Supporting Information. Please add the following statement, or similar, to the Methods: "This study is reported as per the Strengthening the Reporting of Observational Studies in Epidemiology (STROBE) guideline (S1 Checklist)." When completing the checklist, please use section and paragraph numbers, rather than page numbers.

* [FOR POPULATION HEALTH/REGISTRY STUDIES] Please ensure that the study is reported according to the RECORD guideline (available from https://www.record-statement.org) and include the completed checklist as Supporting Information. Please add the following statement, or similar, to the Methods: "This study is reported as per the Reporting of Studies Conducted using Observational Routinely-Collected Data (RECORD) guideline (S1 Checklist)." When completing the checklist, please use section and paragraph numbers, rather than page numbers.

* [FOR POPULATION HEALTH ESTIMATES] Please ensure that the study is reported according to the GATHER statement (available from https://www.equator-network.org/reporting-guidelines/gather-statement) and include the completed checklist as Supporting Information. Please add the following statement, or similar, to the Methods: "This study is reported as per the Guidelines for Accurate and Transparent Health Estimates Reporting (GATHER) statement (S1 Checklist)." When completing the checklist, please use section and paragraph numbers, rather than page numbers.

* [FOR MEDIATION ANALYSES] We recommend that the study is reported according to the AGReMA statement (https://agrema-statement.org/#:~:text=AGReMA%20is%20an%20evidence%2D%20and,randomised%20trials%20and%20observational%20studies) and include the completed checklist as Supporting Information. Please add the following statement, or similar, to the Methods: "This study is reported as per the Guideline for Reporting Mediation Analyses (AGReMA) statement (S1 Checklist)." When completing the checklist, please use section and paragraph numbers, rather than page numbers.

* For all observational studies, in the manuscript text, please indicate: (1) the specific hypotheses you intended to test, (2) the analytical methods by which you planned to test them, (3) the analyses you actually performed, and (4) when reported analyses differ from those that were planned, transparent explanations for differences that affect the reliability of the study's results. If a reported analysis was performed based on an interesting but unanticipated pattern in the data, please be clear that the analysis was data driven.

* Please state in the Methods section whether the study had a prospective protocol or analysis plan. If a prospective analysis plan (from your funding proposal, IRB or other ethics committee submission, study protocol, or other planning document written before analyzing the data) was used in designing the study, please include the relevant document(s) with your revised manuscript as a Supporting Information file to be published alongside your study and cite it in the Methods section. A legend for this file should be included at the end of your manuscript. If no such document exists, please make sure that the Methods section transparently describes when analyses were planned, and when/why any data-driven changes to analyses took place. Changes in the analysis, including those made in response to peer review comments, should be identified as such in the Methods section of the paper, with rationale.

MODELLING STUDIES

The following list is derived from Geoffrey P Garnett, Simon Cousens, Timothy B Hallett, Richard Steketee, Neff Walker. Mathematical models in the evaluation of health programmes. (2011) Lancet DOI:10.1016/S0140-6736(10)61505-X:

* If pertinent, please provide a diagram that shows the model structure, including how the natural history of the disease is represented, the process and determinants of disease acquisition, and how the putative intervention could affect the system.

* Please provide a complete list of model parameters, including clear and precise descriptions of the meaning of each parameter, together with the values or ranges for each, with justification or the primary source cited and important caveats about the use of these values noted.

* Please provide a clear statement about how the model was fitted to the data, including goodness-of-fit measure, the numerical algorithm used, which parameter varied, constraints imposed on parameter values, and starting conditions.

* For uncertainty analyses, please state the sources of uncertainties quantified and not quantified [can include parameter, data, and model structure].

* Please provide sensitivity analyses to identify which parameter values are most important in the model. Uncertainty estimates seek to derive a range of credible results on the basis of an exploration of the range of reasonable parameter values. The choice of method should be presented and justified.

* Please discuss the scientific rationale for the choice of model structure and identify points where this choice could influence conclusions drawn. Please also describe the strength of the scientific basis underlying the key model assumptions.

* For studies that develop a prediction model or evaluate its performance, please ensure that the study is reported according to the TRIPOD statement (https://www.equator-network.org/reporting-guidelines/tripod-statement) and include the completed checklist as Supporting Information. Please add the following statement, or similar, to the Methods: "This study is reported as per the Transparent Reporting of a Multivariable Prediction Model for Individual Prognosis Or Diagnosis (TRIPOD) statement (S1 Checklist)." For studies using machine learning, please use the TRIPOD-AI checklist. When completing the checklist, please use section and paragraph numbers, rather than page numbers.

DIAGNOSTIC STUDIES

* Please ensure that the study is reported according to the STARD guideline (https://www.equator-network.org/reporting-guidelines/stard/) and include the completed STARD checklist as Supporting Information. Please add the following statement, or similar, to the Methods: "This study is reported as per the Standards for Reporting of Diagnostic Accuracy (STARD) guideline (S1 Checklist)." When completing the checklist, please use section and paragraph numbers, rather than page numbers.

* Please structure your Abstract according to STARD for Abstracts (https://www.equator-network.org/reporting-guidelines/stard-abstracts/).

* Please structure the Methods section using the following sub-headings: Study design, Participants, Test methods, Analysis.

* Please include a diagram to describe the flow of participants through the study (typically figure 1).

MENDELIAN RANDOMIZATION STUDIES

* Please ensure that the study is reported according to the STROBE-MR guideline (https://www.equator-network.org/reporting-guidelines/strobe/) and include the completed STROBE-MR checklist as Supporting Information. Please add the following statement, or similar, to the Methods: "This study is reported as per the Strengthening the Reporting of Observational Studies in Epidemiology (STROBE) guideline, specific for mendelian randomization (S1 Checklist)." When completing the checklist, please use section and paragraph numbers, rather than page numbers.

* In the Introduction, please describe the exposure and the evidence for a potential causal relationship between exposure and outcome.

* In the Methods, please explicitly state the 3 core instrumental variable assumptions for the main analysis (relevance, independence, and exclusion restriction), as well assumptions for any additional or sensitivity analysis.

* In the Methods, please describe the MR estimator (e.g., 2-stage least squares, Wald ratio) and related statistics. Detail the included covariates and, in case of 2-sample MR, whether the same covariate set was used for adjustment in the 2 samples.

* If you are presenting an instrumental variable estimate, please compare this to the conventional observational estimate.

* Report the associations between genetic variant and exposure and between genetic variant and outcome, preferably on an interpretable scale.

* Report MR estimates of the relationship between exposure and outcome and the measures of uncertainty from the MR analysis, on an interpretable scale, such as odds ratio or relative risk per SD difference.

* If relevant, please consider translating estimates of relative risk into absolute risk for a meaningful time period.

* Please consider including plots to visualize results (e.g., forest plot, scatterplot of associations between genetic variants and outcome vs between genetic variants and exposure).

SURVEY-BASED STUDIES

* Please ensure that the study is reported according to the CROSS guideline (https://www.equator-network.org/reporting-guidelines/a-consensus-based-checklist-for-reporting-of-survey-studies-cross/) and include the completed CROSS checklist as Supporting Information. Please add the following statement, or similar, to the Methods: "This study is reported as per A Consensus-Based Checklist for Reporting of Survey Studies (CROSS) guideline (S1 Checklist)." When completing the checklist, please use section and paragraph numbers, rather than page numbers.

* Please report your survey response rates according to AAPOR recommendations (https://aapor.org/standards-and-ethics/best-practices/)

* Please define how the population surveyed was sampled.

* Please compare characteristics of respondents and nonrespondents if possible.

* If sequential waves of the survey were sent, please specify whether the characteristics of respondents changed over time or remained constant.

* Please include the survey response rate in the Abstract.

* Please include a copy of the survey in the supplementary files.

SYSTEMATIC REVIEWS & META-ANALYSES

* Please report your SR/MA according to the PRISMA guidelines provided at the EQUATOR site. http://www.equator-network.org/reporting-guidelines/prisma/. Please provide the completed PRISMA checklist as Supporting Information. When completing the checklist, please use section and paragraph numbers, rather than page numbers. Please add the following statement, or similar, to the Methods: "This study is reported as per the Preferred Reporting Items for Systematic Reviews and Meta-Analyses (PRISMA) guideline (S1 Checklist)."

* Abstract: Please report your abstract according to PRISMA for abstracts (https://doi.org/10.1371/journal.pmed.1001419) following the PLOS Medicine abstract structure (Background, Methods and Findings, Conclusions). Please ensure you provide dates of search, data sources, number of studies included, types of study designs included, eligibility criteria, and synthesis/appraisal methods.

* Please note that we expect searches to be updated to within 6 months of the time of submission.

QUALITATIVE STUDIES

* Please report your qualitative study according to the appropriate study design provided at (http://www.equator-network.org/?post_type=eq_guidelines&eq_guidelines_study_design=qualitative-research&eq_guidelines_clinical_specialty=0&eq_guidelines_report_section=0&s=) and provide the relevant completed checklist as a supplemental file. In the checklist, please include sufficient text excerpted from the manuscript to explain how you accomplished all applicable items. When completing checklists, please use section and paragraph numbers, rather than page numbers.

* We recommend that authors use the COREQ checklist, or other relevant checklists listed by the Equator Network, such as the SRQR, to ensure complete reporting (see: http://www.equator-network.org/?post_type=eq_guidelines&eq_guidelines_study_design=qualitative-research&eq_guidelines_clinical_specialty=0&eq_guidelines_report_section=0&s=). Please add the following statement, or similar, to the Methods: "This study is reported as per the Consolidated criteria for reporting qualitative research (COREQ): a 32-item checklist for interviews and focus groups (S1 Checklist)."

* In general, we expect qualitative studies to include the following: 1) defined objectives or research questions; 2) description of the sampling strategy, including rationale for the recruitment method, participant inclusion/exclusion criteria and the number of participants recruited; 3) detailed reporting of the data collection procedures; 4) data analysis procedures described in sufficient detail to enable replication; 5) a discussion of potential sources of bias; and 6) a discussion of limitations.

HEALTH ECONOMICS / COST-EFFECTIVENESS STUDIES

* Please ensure that the study is reported according to the CHEERS guideline (available from: https://www.equator-network.org/reporting-guidelines/cheers) and include the completed checklist as Supporting Information. Please add the following statement, or similar, to the Methods: "This study is reported as per the Strengthening the Consolidated Health Economic Evaluation Reporting Standards 2022 (CHEERS 2022) Statement (S1 Checklist)." When completing the checklist, please use section and paragraph numbers, rather than page numbers.

---

## [Decision Letter · Decision Letter 3]

16 Apr 2025

Dear Dr. Shi,

Thank you very much for re-submitting your manuscript "Natural cycle versus hormone replacement therapy as endometrial preparation in ovulatory women undergoing frozen-thawed embryo transfer: the COMPETE open-label randomized controlled trial" (PMEDICINE-D-24-02242R3) for review by PLOS Medicine.

I have discussed the paper with my colleagues and it was also seen again by 3 reviewers. I am pleased to say that provided the remaining editorial and production issues are dealt with we are planning to accept the paper for publication in the journal.

[LINK]

We look forward to receiving the revised manuscript by Apr 23 2025 11:59PM.

Sincerely,

Alison Farrell, Ph.D.

Senior Editor

PLOS Medicine

plosmedicine.org

Requests from Editors:

* In the author summary, in the final bullet point of 'What Do These Findings Mean?', please include the main limitations of the study in non-technical language.

* PLOS’ policy on data availability specifies that data must be handled so as to not compromise study participants’ privacy. Authors must follow applicable laws in ensuring they do not compromise participant privacy (http://journals.plos.org/plosone/s/data-availability). This includes the requirements under GDPR.

Where an author as part of research gathers personally-identifying information (“PII”) on EU citizens, the author should make the disclosures to and obtain the consent(s) from the subjects as required under GDPR https://bit.ly/2Fa05Kl.

Manuscripts submitted to PLOS should not contain research participants personally-identifying information. In rare exceptions where this is unavoidable and a manuscript does contain PII, the authors should be willing and able to provide PLOS with confirmation of GDPR compliance upon request.

Please confirm that your study data complies with PLOS' policy.

* Please include the statement on code availability in the data availability statement. We cannot open the code document. Please deposit new code in a public repository and include the relevant link.

* When completing the CONSORT checklist, please use section and paragraph numbers, rather than page numbers. The checklist should be included as supporting information, and should be cited in the article. Please use the CONSORT 2025 checklist.

* If your trial had to undergo important modifications in response to extenuating circumstances, please complete the CONSERVE-CONSORT checklist and provide in your Supporting Information.

* Please state in the Methods any amendments to the protocol and provide the amended protocol for inclusion in supplementary information.

* Clearly signpost any exploratory and post hoc analyses.

Line edits:

line 51--birth rate?

line 59--capture method?

line 56--in 'a' single assisted reproduction centre....

line 99--remove the statement "because it lowers". This study does not show active lowering by HRT. Please temper conclusions and rephrase as associations instead.

line 102--temper this statement and add a bullet point on limitations.

line 475--'for' whom

line 516--find a synonym for 'empowers' to avoid repetition

Comments from Reviewers:

Reviewer #1: I have no further comments - my previous queries have been fully addressed by the authors in the current revision.

Reviewer #3: I think the authors have taken into consideration all relevant comments. Thank you. Congratulations.

Reviewer #5: The responses to reviewer 5 comments are acceptable to me.

[LINK]

---

## [Editor Report · Decision Letter 4]

7 May 2025

Dear Dr Shi,

On behalf of my colleagues and the Academic Editor, Elizabeth Ginsburg, I am pleased to inform you that we have agreed to publish your manuscript "Natural cycle versus hormone replacement therapy as endometrial preparation in ovulatory women undergoing frozen-thawed embryo transfer: the COMPETE open-label randomized controlled trial" (PMEDICINE-D-24-02242R4) in PLOS Medicine.

PRESS

Sincerely,

Alison Farrell, Ph.D.

Senior Editor

PLOS Medicine